# Co-regulation of the transcription controlling ATF2 phosphoswitch by JNK and p38

Klára Kirsch[1], András Zeke [1], Orsolya Tőke[2], Péter Sok [1], Ashish Sethi [3], Anna Sebő[1], Ganesan Senthil Kumar[4], Péter Egri[1], Ádám L. Póti[1], Paul Gooley [3], Wolfgang Peti [4], Isabel Bento[5], Anita Alexa[1] & Attila Reményi [1✉]

Transcription factor phosphorylation at specific sites often activates gene expression, but how environmental cues quantitatively control transcription is not well-understood. Activating protein 1 transcription factors are phosphorylated by mitogen-activated protein kinases (MAPK) in their transactivation domains (TAD) at so-called phosphoswitches, which are a hallmark in response to growth factors, cytokines or stress. We show that the ATF2 TAD is controlled by functionally distinct signaling pathways (JNK and p38) through structurally different MAPK binding sites. Moreover, JNK mediated phosphorylation at an evolutionarily more recent site diminishes p38 binding and made the phosphoswitch differently sensitive to JNK and p38 in vertebrates. Structures of MAPK-TAD complexes and mechanistic modeling of ATF2 TAD phosphorylation in cells suggest that kinase binding motifs and phosphorylation sites line up to maximize MAPK based co-regulation. This study shows how the activity of an ancient transcription controlling phosphoswitch became dependent on the relative flux of upstream signals.

[1] Biomolecular Interactions Research Group, Institute of Organic Chemistry, Research Center for Natural Sciences, H-1117 Budapest, Hungary. [2] Laboratory for NMR Spectroscopy, Research Center for Natural Sciences, H-1117 Budapest, Hungary. [3] Department of Biochemistry and Molecular Biology and Bio21 Molecular Science and Biotechnology Institute, University of Melbourne, Parkville, VIC 3010, Australia. [4] Department of Chemistry and Biochemistry, University of Arizona, Tucson, AZ, USA. [5] European Molecular Biology Laboratory, Hamburg, Germany. ✉email: remenyi.attila@ttk.hu

Receptor activation by extracellular cues at the cell membrane and gene expression in the nucleus are linked through signaling pathways involving protein kinases. To reveal the molecular basis underlying the transcriptional response of cells to complex biological inputs, it is important to explore how protein kinases control transcription factors (TF). TFs bind to DNA sites using their DNA-binding domain (DBD) and to cofactors using their transcription activating domain (TAD). TADs are often unstructured and enable regulated binding to cofactors by using phosphoswitches: regions functionally affected by protein phosphorylation and evolved to respond to post-translational modification[1,2]. TAD phosphorylation is a major mechanism through which environmental cues affect cell physiology[3].

Mitogen-activated protein kinases (MAPK) control transcription by phosphorylating TADs. For example, ERK1/2 phosphorylates Elk1—an immediate-early (IE) transcription factor—on its TAD promoting its interaction with a specific subunit of the RNA II polymerase binding Mediator complex involved in transcription initiation[4,5]. Similarly, activator protein 1 (AP-1) transcription factors, such as c-Fos, c-Jun or ATFs, contain MAPK regulated phosphosites in their TAD[6–8]. Ser63/Thr73 from human c-Jun or Thr69/71 from ATF2 are known to be phosphorylated by MAPKs leading to upregulation of AP-1 transcriptional activity[9,10]. The ATF2 TAD was analyzed by NMR spectroscopy, and apart from a small C2H2-type Zn-finger, it was found to be disordered[11]. Phosphosites in AP-1 TFs are essential in making phosphorylation-dependent contacts with gene expression controlling cofactors/effectors (e.g., p300/CBP—a coactivator/adaptor protein with histone acetyltransferase activity—is an important ATF2 cofactor[12]), but how AP-1 TAD phosphorylation leads to increased transcription is not yet understood.

Activating transcription factor 2 (ATF2) is ubiquitously expressed in most cell types and regulates the transcription of various genes involved in a broad spectrum of cellular functions: cell growth, development and stress[13]. JNK and p38 were shown to control ATF2 function via the 69-TPTP-72 phosphoswitch[10,14]. MAPKs phosphorylate their substrates on Ser/Thr-Pro target motifs (S/TP) but substrate phosphorylation requires binding to at least one of two distinct protein–protein interaction hot-spots on the MAPK: the D- or the F-groove[15]. The MAPK D-groove (also referred to as the D-motif recruitment site, DRS) binds to D(ocking)-motifs comprising a positively charged and a hydrophobic region interacting with the negatively charged common docking (CD) groove and the hydrophobic pockets, respectively[16]. The F-groove (also referred to as the F-motif recruitment site, FRS) binds F(xFP)-type motifs and is located at the opposite side of the kinase compared to the D-groove. F-peptides bind to the activated, double-phosphorylated ppMAPK with increased affinity[17,18]. D-motifs and F-motifs are differently located compared to the MAPK phosphorylation target site (S/TP): 10-50 amino acids N-terminal or 10-20 amino acids C-terminal, respectively[19].

MAPKs play central roles in many signaling pathways and all three major groups of them (ERK1/2, JNK, and p38) are ubiquitously expressed. Their activation depends on the cell-type and the nature of the input signal, however, most physiological stimuli activate more than one MAPK, which brings about systems level opportunities for more complex regulation. What is the mechanistic basis of this regulation operating at the MAPK-TF level? How do TADs contribute to quantitative gene expression depending on the strength of upstream signals?

In the present study, we explore the structural and mechanistic basis of ATF2 regulation by two MAPKs: JNK and p38. These two MAPKs have distinct physiological roles and their activity is often altered in human tumors and cancer cell lines[20]. We started by mapping out the ATF2 TAD region that binds JNK. We find that a Zn-finger located next to a D-motif is required to mediate MAPK-specific binding. Next, we identify a region that binds the phosphorylated form of p38 selectively. MAPK-ATF2 TAD complexes are structurally elucidated by X-ray crystallography/NMR and monitored by protein–protein interaction assays in cells. These data are used to create a mechanistic model of cellular TAD phosphorylation that reveals how two distinct signaling pathways converge on the same transcription controlling phosphoswitch and enable ATF2 TAD mediated quantitative gene expression.

## Results

**JNK binds to ATF2 using a linear D-motif and a Zn-finger.** The transactivating region of both c-Jun and ATF2 contains a MAPK binding D-motif. The structure of the JNK1-ATF2 D-motif peptide complex was earlier determined by X-ray crystallography[21]. In contrast to c-Jun, which binds JNK using a D-motif only, ATF proteins (e.g., ATF2, ATF7, and CREB5) contain a C2H2 Zn-finger which partially overlaps with the adjacent D-motif (Supplementary Fig. 1a–c). This proximal arrangement between a structured domain and the D-motif consensus sequence seems to be an evolutionarily conserved trait in ATFs, suggesting that JNK-ATF binding needs to be studied using protein constructs including both regions.

In order to determine the minimal binding region, we probed the full transactivation region (ATF2 residues 19–100) by GST pull-down assays using purified JNK1 (Fig. 1a). This showed that neither the Zn-finger (19–50) nor the D-motif (42–55) alone was sufficient for binding and that the minimal region (19–58) contained both the Zn-finger and the D-motif. This construct bound JNK with micromolar affinity (~5 μM) and the interaction required an intact Zn-finger, as a chelating agent (EDTA) diminished binding (Fig. 1b,c). The interaction also depended on the basic region of the D-motif, as a charge reversal mutation (K48E) also diminished binding (Fig. 1b). In contrast, the JNK-c-Jun interaction was not sensitive to the presence of EDTA, but a charge reversal mutation in ATF2 or in c-Jun (K48E or K35E, respectively) decreased binding similarly (~10 fold) (Fig. 1d). Isothermal titration calorimetry (ITC) also showed an ~10 fold decrease in JNK-ATF2 binding in the presence of EDTA, due to an increase in the entropic cost of binding as the enthalpic contribution stayed nearly the same (Fig. 1e). This suggests that an intact Zn-finger is necessary for binding by possibly constraining the flexibility of critical D-motif residues.

Next, we assessed the role of JNK binding on MAPK target site phosphorylation. The Zn-finger + D-motif module (19–58) was left intact or mutated (K48E) or replaced by c-Jun or its mutated version (K35E) and phosphorylation of the 69-TPTP-72 target site by activated JNK was monitored by anti-phospho immunoblots (Fig. 1f). This analysis confirmed the importance of key lysine residues in ATF2 Zn-finger + D-motif or c-Jun D-motif mediated phosphorylation.

The importance of the Zn-finger + D-motif module in JNK binding, full-length ATF2 phosphorylation, and ATF2 TAD mediated transcriptional activity was also examined in HEK293T cells (Supplementary Fig. 2). The K48E charge reversal mutation decreased full-length ATF2-JNK1 binding in a yellow fluorescent protein (YFP) fragment complementation experiment and Zn-finger disrupting mutations (C27A and C32A; CACA) had the same effect (Supplementary Fig. 2a). The YFP signal was mostly localized in the cytosol, but co-expression with full-length c-Jun shifted the fluorescent signal from the cytosol to the nucleus (Supplementary Fig. 2b). This is in agreement with earlier findings demonstrating that nuclear translocation of AP-1

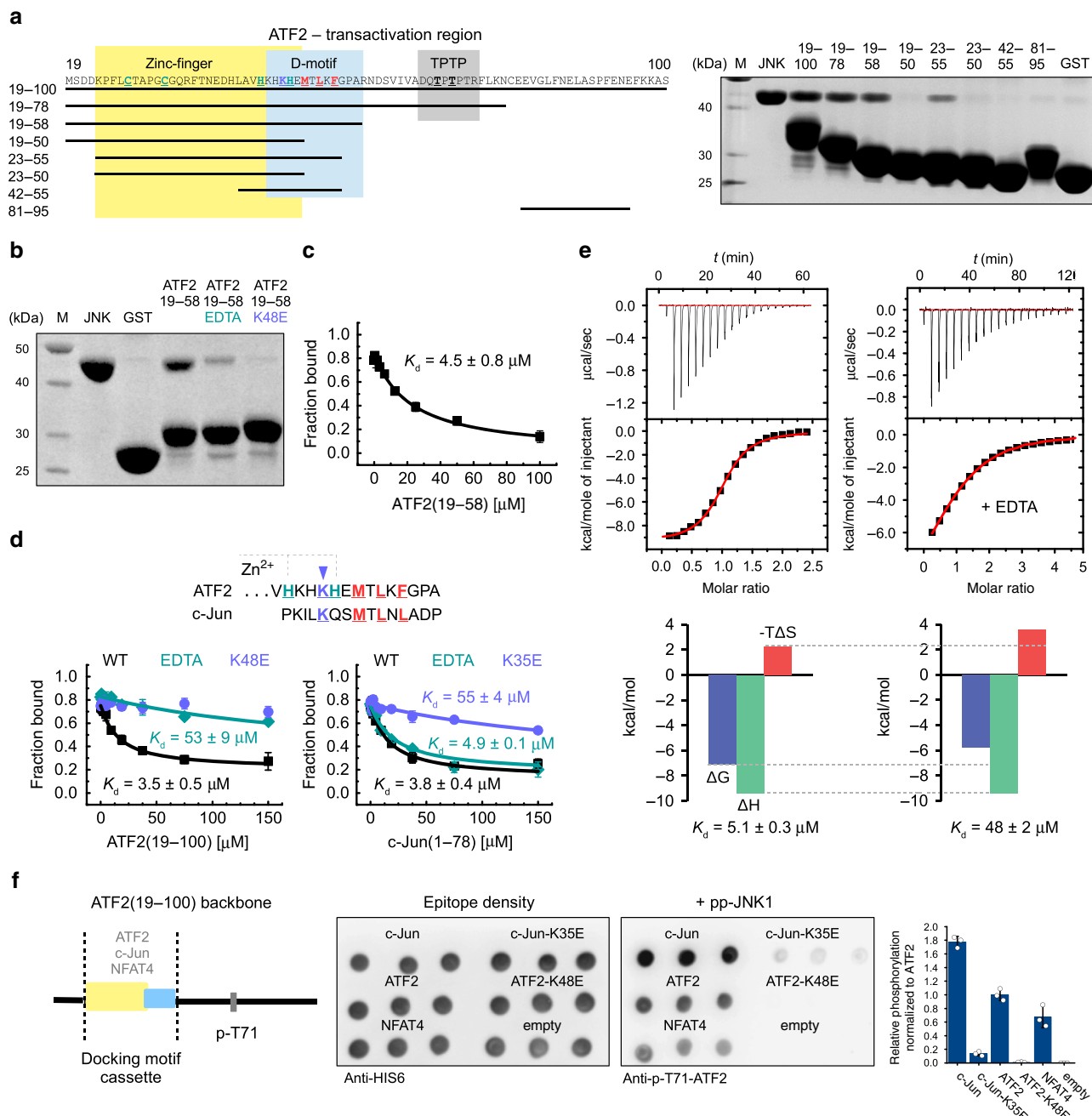

**Fig. 1 JNK-ATF2 binding depends on an intact Zn-finger + D-motif module. a** Mapping of JNK1 binding to ATF2 by GST pull-down. Interaction of recombinant expressed and purified bait and prey was detected on Coomassie stained SDS-PAGE. GST control was used as negative control to assess unspecific binding of the prey (GST; M—molecular weight marker, JNK—load). The experiment was repeated independently at least three times with similar results. **b** GST pull-down experiment with JNK1 (prey) and GST-ATF2(19–58) (bait). The binding mix contained 2 mM EDTA or the bait had a lysine to glutamate mutation (K48E). **c** Result of a competitive fluorescent polarization-based binding assay. Error bars on the binding isotherms show SD based on three independent measurements ($n = 3$). Data are represented as mean values ± SD. **d** Binding of JNK1 to MBP-ATF2(19–100) and MBP-c-Jun(1–78) fusion proteins in the presence of EDTA, or when lysine residues (K48E or K35E in ATF2 and c-Jun, respectively) were mutated. Error bars on the binding isotherms show SD based on three independent measurements ($n = 3$). Date are represented as mean values ± SD. (MBP: maltose-binding protein fusion tag.) **e** ITC measurements of JNK1 and ATF2(19–58) binding in the absence or presence of 2 mM EDTA. Representative ITC data (upper panels) are shown with raw injection heats and the corresponding specific binding isotherms. Lower panels show the enthalpic ($\Delta H$) and entropic ($-T\Delta S$) contributions to binding free energy ($\Delta G$). **f** Phosphorylation of ATF2 reporter constructs. GST constructs containing different docking motif cassettes were dotted on a membrane and incubated with activated JNK1 (pp-JNK1). Equal load was confirmed by Anti-His6 antibody, while phosphorylation of the probe was determined by Anti-p-T71-ATF2 immunoblots. (empty: no docking motif; NFAT4: the D-motif from NFAT4 was used as reference to former work[16]). Error bars show SD from three technical replicates. Data points are represented as mean values ± SD. The experiment was repeated twice with similar results. Source data are provided as a Source data file.

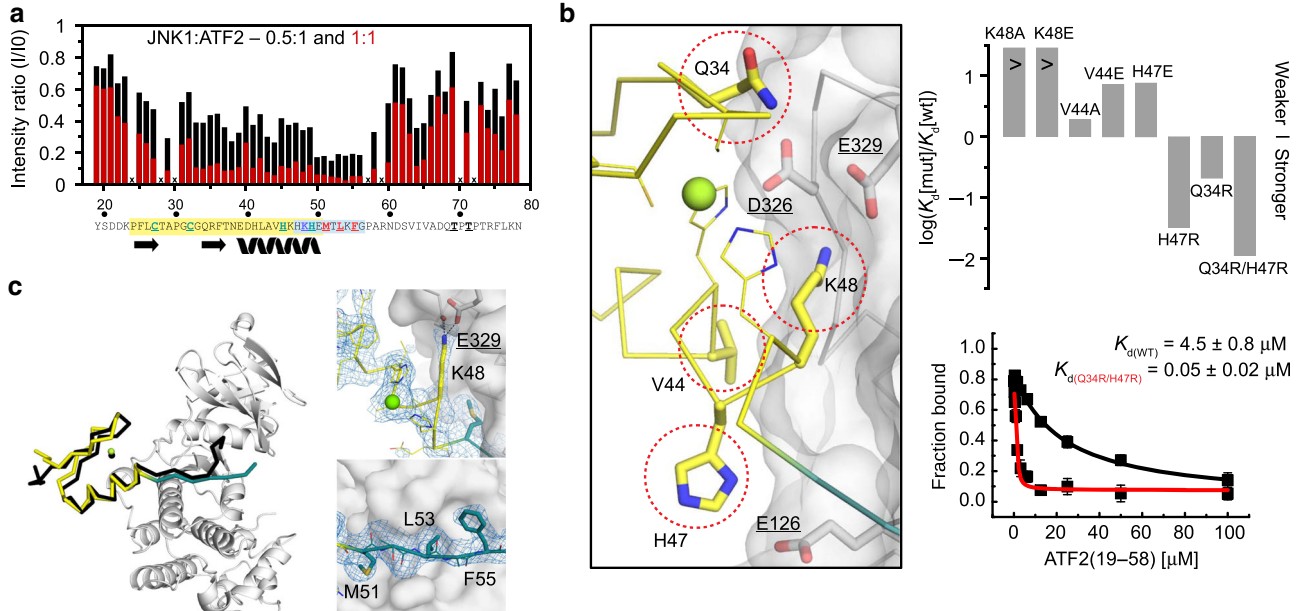

**Fig. 2 Structure of the JNK1-ATF2 complex. a** Interaction region mapping by NMR. [15]N-labeled ATF2(19-78) was mixed with JNK1 in two different molar ratio, and intensity changes, normalized to the peak intensity without JNK1, for ATF2 [1]H-[15]N HSQC peaks for the 0.5:1 (black) or 1:1 (red) JNK1:ATF2 experiment were plotted. 'x' indicates resonances of residues which are not resolved in the spectra and, therefore, could not be quantified or prolines. **b** Analysis of the JNK-ATF2(Zn-finger) protein–protein binding interface based on the HADDOCK model. Amino acid residues lying at the interface (Q34, V44, H47, and K48) were mutated to alanine, glutamate or arginine. The binding affinity of mutant ATF2(19–100) constructs (mut) were compared to that of the wild-type (wt) (Supplementary Fig. 4). The panels on the right show the summary of competitive fluorescence polarization based protein–protein binding assay results on top and the competitive binding curves for the wild-type (black) and for the Q34R/H47R double mutant (red) below. Error bars on the binding isotherms show SD based on three independent measurements ($n = 3$). Date are represented as mean values ± SD. **c** Crystal structure of the JNK1-ATF2(19–58) complex. The left panel shows the superposition of the HADDOCK model (black) with the crystal structure (yellow+cyan; the RMSD of Cα atoms for the ATF2(24-55) region using JNK as the reference, was 0.9 Å). Panels on the right show zoomed-in views at the CD-groove highlighting K48 (top) or at the hydrophobic groove highlighting M51, L53, and F55 (bottom) from the ATF2 D-motif. The simulated annealing Fo-Fc omit map of ATF2 is shown at 1.5σ. Source data are provided as a Source data file.

transcription factors may be controlled by c-Jun/ATF2 hetero-dimerization[22]. JNK-specific phosphorylation of full-length ATF2 was examined in an engineered HEK293T cell line (HT-MLK3-MKK7) in which MAPK activity was turned on by inducible expression of specific upstream activators of JNK. These experiments showed that full-length ATF2 is less phosphorylated upon JNK activation when the critical residues (K48, CACA) are mutated (Supplementary Fig. 2c). In addition, a GAL4 promoter/luciferase assay confirmed the role of these residues in ATF2 TAD mediated basal transcription (Supplementary Fig. 2d).

**Structure of the JNK1-ATF2 complex.** We examined JNK1-ATF2 binding by NMR spectroscopy based on an existing NMR spectral assignment and structure for ATF2 and crystal structure of JNK-peptide interactions. Different amounts of JNK1 was added to [15]N-labeled ATF2(19-78), and the intensity of HN cross-peaks in 2D [[1]H,[15]N] HSQC spectra was monitored (Fig. 2a and Supplementary Fig. 3a). In addition to the expected changes in the ATF2 D-motif, the intensity of Zn-finger residues also similarly changed confirming that this small domain contributes to binding. Next, we created a structural model of the JNK-ATF2 (19–58) complex using the High Ambiguity Driven protein Docking (HADDOCK) approach to see how the Zn-finger complements the D-motif in JNK binding[23] (Supplementary Fig. 3b,c). In addition, we set out to determine the crystal structure of the JNK-ATF2 complex using the ATF2(19–58)_Q34R/H47R construct that bound to JNK1 ~100-fold stronger compared to wild-type (Fig. 2b). The crystal structure showed that M51, L53, and F55 from the ATF2 D-motif bind in the

hydrophobic pockets, while K48 binds in the negatively charged CD groove (Fig. 2c). Zn-finger residues also make contacts to JNK1 and mutations of these (K48, V44, H47 or Q34) greatly affected binding. Furthermore, arginine replacements at position 34 and 47 increased binding by creating additional H-bonds at the JNK:Zn-finger interface (Supplementary Figs. 3d and 4). The new crystallographic model provides the structure of the biologically relevant JNK1-ATF2 complex (Supplementary Fig. 3e).

**p38 and ATF2 TAD binding.** In addition to JNK, p38 also regulates ATF2 and thus we wanted to test if the interaction is similar or different. We used a luciferase complementation based dynamic protein–protein interaction (PPI) assay (NanoBit) to monitor both p38α and JNK1 binding to ATF2 TAD in HEK293T cells. Cells were treated with anisomycin, which activates both JNK and p38[24,25] (Fig. 3a). JNK-ATF2 binding was found to be the same if activated or not, but p38-ATF2 binding was elevated when p38 was activated. To map the p38 binding region of ATF2, phosphorylation of different ATF2 TAD constructs by activated p38α (pp-p38) was tested (Supplementary Fig. 5a). This analysis unexpectedly identified a critical short sequence (92-FENEF-96) C-terminal to the 69-TPTP-72 phosphoswitch. In ATF2 variants lacking the two phenylalanine residues (92-FENEF-96 → 92-AENEA-96; referred to as MUT4), phosphorylation of the TAD by pp-p38 was greatly reduced, but JNK mediated phosphorylation remained unaffected.

The role of all evolutionarily conserved phenylalanine residues C-terminal to the phosphoswitch was then tested (MUT1, 2, 3, and 4) in a HEK293 cell line engineered for MAPK-specific

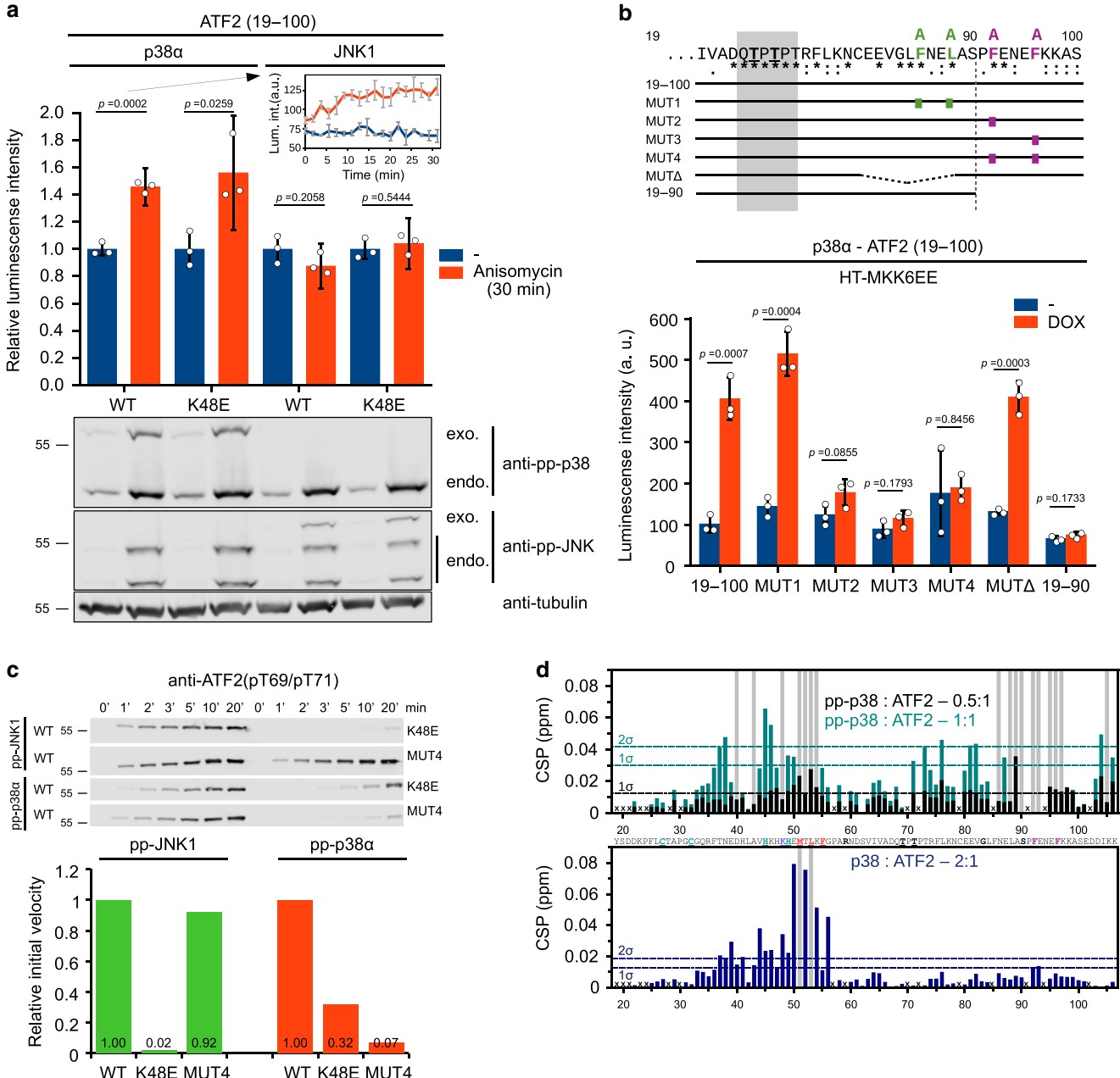

**Fig. 3 p38 mediated binding and phosphorylation of ATF2 TAD. a** Luciferase complementation assay (NanoBit) on p38-ATF2 and JNK1-ATF2 binding in HEK293T cells. The impact of p38 and JNK activation by anisomycin and the K48E mutation were examined. Luminescence signal change upon MAPK activation was monitored in time after addition of anisomycin (see inset for p38:WT-ATF2 binding). Complex formation between p38α or JNK1 and ATF2 (19–100) was measured after the addition of the luciferase substrate Coelenterazine h (at 30 min) and luminescence values obtained in the presence of anisomycin (red) were normalized to control (−; blue). MAPK activation was confirmed by anti-phosphoMAPK (anti-pp-p38 or anti-pp-JNK) western-blots. (endo: endogenous MAPKs; exo: luciferase fragment tagged MAPK probe). Anti-tubulin antibody was used to demonstrate equal sample load. Error bars show SD calculated based on three technical replicates. Bar charts show data points with mean ± SD and p-value, two-sided unpaired t-test. **b** Mapping critical amino acid residues for pp-p38-ATF2 TAD binding in HEK293T cells. The luciferase complementation based NanoBit assay was used to test the binding of p38 and different ATF2 TAD (19–100) mutants. Selective activation of p38 was achieved by doxycycline (DOX) addition in HT-MKK6-EE cells: DOX induces the expression of MKK6EE, which is the upstream activator kinase of p38. Error bars show SD calculated based on three technical replicates. Bar charts show data points with mean ± SD and p-value, two-sided unpaired t-test. **c** In vitro phosphorylation of ATF2 TAD constructs. JNK1 or p38α mediated phosphorylation of two ATF2 TAD (19–100) constructs (K48E and MUT4) were compared to wild-type (WT). ATF2 TAD phosphorylation was detected by using anti-phosphoT69/71 antibody in western-blots. **d** Interaction region mapping by NMR. [15]N-labeled ATF2(19–106; 100 μM) was mixed with pp-p38 (upper panel) or nonphosphorylated p38 (lower panel) in different molar ratio. Histograms show the [1]H/[15]N chemical shift perturbations (CSPs) vs residue number (for intensity changes upon binding see Supplementary Fig. 5b). Dashed lines indicate CSPs corresponding to 1σ or 2σ changes. Residues displaying line broadening are indicated by gray bars. 'x' indicates resonances of residues that were not resolved in the spectra and therefore could not be quantified or prolines. Source data are provided as a Source data file.

activation (HT-M6) (Fig. 3b). Specific activation of p38, but not that of JNK or other MAPKs, was achieved by Tet-on inducible expression of the p38-specific MAPK kinase (MKK6) with doxycycline[26]. These experiments confirmed the pivotal role of two phenylalanine residues (92-FENEF-96) in binding to pp-p38. To assess the importance of the Zn-finger + D-motif module and the newly identified FENEF region, we monitored the phosphorylation of mutated TAD constructs (K48E and MUT4) by pp-JNK1 and pp-p38 in vitro (Fig. 3c). These experiments demonstrated that both the Zn-finger+D-motif module and the FENEF motif are indispensable for p38 mediated TAD phosphorylation. In contrast, JNK mediated phosphorylation was unaffected by FENEF motif mutations but was greatly impaired by the K48E mutation.

Next, p38 binding of ATF2 TAD was explored by using NMR spectroscopy. ATF2 (19-106) was $^{15}$N-labeled, mixed with pp-p38 or nonphosphorylated (np) p38, and chemical shift perturbations (CSP) and intensity of HN cross-peaks in 2D [$^{1}$H,$^{15}$N] HSQC spectra were compared (Fig. 3d and Supplementary Fig. 5b). Residues displaying line broadening or large CSPs mapped to the FENEF motif and also to the D-motif with pp-p38, while np-p38 affected only the D-motif region weakly. This analysis showed that the major interaction site for pp-p38 is at the FENEF motif but some binding to the D-motif region was also detected.

**JNK phosphorylation attenuates p38-SPFENEF motif binding**. The region between the phosphoswitch and the FENEF motif in ATF2 contains a putative S/TP phosphorylation site (S90). This SP site is only conserved in vertebrates, but not in other eukaryotes: the serine is replaced by asparagine in all invertebrate ATF2 orthologs (see Supplementary Fig. 1c). Interestingly, Ser90 was shown to be phosphorylated only by JNK but not by p38[27]. Intrigued by these observations, we created a human ATF2 TAD construct in which Ser90 was changed to asparagine (S90N) and analyzed MAPK-ATF2 TAD binding in HEK293T cells using the NanoBit PPI assay. We used anisomycin treatment to activate JNK and p38 in parallel (Fig. 4a). ATF2 S90N showed increased binding over WT-ATF2 to pp-p38 and p38 binding was not affected by JNK phosphorylation. In contrast, JNK activation decreased binding of ATF2 TAD to pp-p38, since cells in the presence of a JNK-specific inhibitor (JNK-IN-8) displayed elevated pp-p38:WT TAD binding. Next, the effect of chemically synthesized peptides (83-GLFNELA**S**PFENEFKKASED-102, referred to as SPFENEF peptide henceforth) on pp-p38 mediated ATF2 phosphorylation was tested in vitro. Three peptides (WT, S90N and pS90) added in trans inhibited phosphoswitch phosphorylation differently in vitro ($K_i$ = 33.6 µM, 7.5 µM and 820 µM, respectively) (Fig. 4b). These results showed that N90 promotes pp-p38:ATF2-TAD binding, while Ser90 phosphorylation by JNK in vertebrate orthologs negatively affects it (Fig. 4c).

**Structure of the p38-SPFENEF motif complex**. After mapping out the p38 interacting regions of the ATF2 TAD, we wanted to explore the binding surface of the SPFENEF motif on pp-p38. First, we used solution state biomolecular NMR spectroscopy. We compared the 2D [$^{1}$H,$^{15}$N] TROSY spectra of ($^{2}$H,$^{15}$N)-labeled np-p38 or pp-p38 when titrated with WT or the S90N SPFENEF peptides (ATF2 83–102). Since we previously determined the sequence-specific backbone assignment of np/pp-p38, we were able to identify those p38 resonances that showed significant intensity change upon binding the SPFENEF motif[28]. Interestingly, the interaction is in slow exchange in the NMR timescale—where the chemical shifts for bound and free pp-p38 could be both observed. This is in contrast to other weak MAPK-protein

complexes that normally display fast or intermediate exchange, where peaks shift or undergo large intensity changes, respectively. In brief, the doubling of peaks suggests that the ATF2 TAD-bound pp-p38 state is unique and differs form the unbound state. We found that a great number of pp-p38 peaks were affected, but strikingly np-p38 did not show any major changes (Supplementary Fig. 6a). This confirms that SPFENEF specifically binds to pp-p38. Notably, pp-p38 residues at the FRS displayed significant intensity changes in the presence of S/NPFENEF peptides. We also determined the crystal structure of the pp-p38:ATF2(83–102)_S90N complex (Fig. 5a and Supplementary Table 1). This structure revealed that the FENEF motif adopts an α-helix which is wedged in-between the N-terminal half of αG and one of the α-helices from the so-called MAPK insert (α2L14). The bulky hydrophobic side-chains of F92 and F96 are buried at the FRS pocket that forms only in pp-p38 when the activation loop is double-phosphorylated (Fig. 5b) and W197 (located in the αEF/αF loop) is flipped out from its original position (Fig. 5c). This explains why ATF2 binding to p38 depends on the activation state of the kinase.

Next, we created a HADDOCK model of the pp-p38:ATF2 (TAD) complex that included the Zn-finger + D-motif region docked into the DRS, the phosphoswitch region docked at the MAPK active site, and the SPFENEF peptide docked to the FRS (Fig. 5d). This structure showed that phosphoswitch (69-TPTP-72) binding at the active site is compatible with the bipartite binding of the TAD at the DRS and the FRS, and explained why mutations at either of these regions (e.g., K48E and MUT4 constructs, see Fig. 3c) decreased phosphorylation. The model also hinted on the role of S90 (as well as on the impact of asparagine replacement or phosphorylation at this position) in the regulation of the pp-p38:ATF2 complex: the residue at 90 is placed into a key position where the chemical nature of its side chain affects the conformation of the SPFENEF motif by influencing intra-motif and intermolecular H-bond formation. Apart from direct contacts between pp-p38 and the SPFENEF motif, there are helix capping interactions that stabilize the α-helical conformation of α2L14, αG, and the ATF2 helix itself by involving glutamate (E93, E95) or the S/N90 side-chains. Phosphorylation at S90 is sterically detrimental (Fig. 5e). Moreover, the structural model explains why the asparagine in invertebrates enables higher affinity binding (Fig. 5f).

In addition, we also analyzed the NMR structure of WT, S90N and pS90 peptides in solution. This analysis suggested that this ATF2 TAD region has α-helical propensity (up to ~20% in aqueous buffer) around the critical SP site, but depending on the residue at 90, the conformation of the peptide varied (Supplementary Fig. 6b). In summary, these structural data show how the amino acid at 90 affects the conformation and p38-binding capacity of the SPFENEF motif.

**Transcriptional activation by two MAPKs**. We showed that JNK and p38 bind to the ATF2 TAD through distinct regions. To specifically address the role of these two MAPKs in ATF2 TAD mediated transcription, wild-type TAD and different binding site mutants thereof were expressed as fusion proteins with the GAL4 DNA-binding domain and their capacity to activate GAL4 DNA-binding based transcription were tested. In agreement to the results of in vitro or cell-based binding experiments, all mutations (both in the Zn-finger + D-motif or in the SPFENEF motif), apart from the S90N mutation, showed decreased basal transcription from a luciferase expression based reporter vector (Fig. 6a). Basal transcription from the reporter vector was decreased in the presence of JNK- or p38-specific inhibitors (JNK-IN-8 or SB202190, respectively) or increased by heterologous expression of JNK1 or

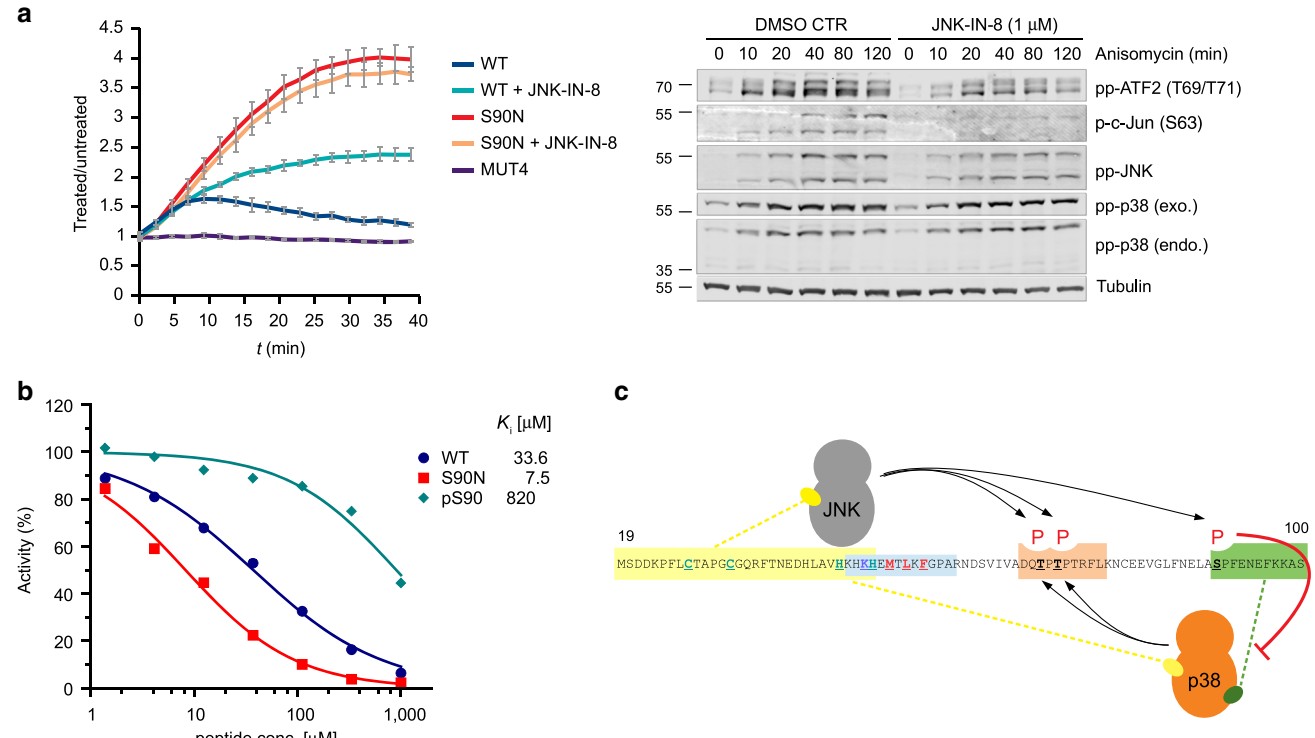

**Fig. 4 JNK mediated phosphorylation affects p38-ATF2 TAD binding. a** p38-ATF2 TAD binary interaction after anisomycin treatment. Binding was monitored in time after the addition of anisomycin using a luciferase complementation based protein–protein binding assay (NanoBit) in live cells (left panel). Anisomycin treatment turns on p38 and JNK pathway activity in HEK293T cells (pp-p38 or pp-JNK—or p-c-Jun—western-blot signal, respectively; right panel; min—minutes). Wild-type (WT), S90N, and FENEF region mutated (MUT4) TAD constructs interacted with the p38 luciferase probe differently after turning on p38 phosphorylation by anisomycin (10μg/mL). Luminescence signal was normalized to cells not treated by anisomycin (treated/untreated). The panel on the right shows results of western-blots with different antibodies; exo: luciferase fragment tagged exogenous p38 probe; endo: endogenous p38; tubulin: anti-tubulin antibody used as the load control. DMSO CTR: cells treated only by anisomycin. JNK-IN-8: cells treated with anisomycin and a JNK-specific inhibitor. MAPK mediated ATF2 TAD phosphorylation at the phosphoswitch was monitored with the pp-ATF2(T69/71) antibody. Error bars show SD from three technical replicates. Data are represented as mean values ± SD. The experiment was repeated twice with similar results. **b** Inhibition of pp-p38 mediated TAD phosphorylation by SPFENEF motif-containing peptides in vitro. WT, S90N and S90 phosphorylated (pS90) peptides (83–102) were added in increasing amounts to an in vitro pp-p38 → TAD(19–100) phosphorylation reaction. The Ki values for the three peptides are highlighted. **c** Schematic of MAPK binding to different ATF2 TAD regions and phosphorylation of critical S/TP sites. Source data are provided as a Source data file.

p38α, but the impact on p38 was only apparent in the case of the S90N construct (Fig. 6b, c).

Next, the ATF2 TAD was turned on by artificial and highly specific induction of JNK or p38. HEK293T cells were transfected with an expression plasmid allowing doxycycline (DOX) inducible (Tet-on) expression of an MLK3-MKK7 fusion construct or a constitutive active version of MKK6 (MKK6) to ensure JNK- or p38-specific activation, respectively. The HT-MLK3-MKK7 and HT-MKK6EE stable cells lines (HT: HEK293T Tet-on) allowed single MAPK activation, but the HT-MKK7-MLK3/MKK6EE cell line allowed controlled double activation of JNK and p38 at the same time (Fig. 6d). These experiments, which lacked the pitfalls of cross-activation of both MAPK pathways by unspecific stimuli (e.g., anisomycin or UV radiation), demonstrated that under these conditions JNK and p38 mediated activation of the ATF2 TAD can indeed be synergistic.

**Quantitative modeling of MAPK mediated ATF2 TAD phosphorylation**. To simulate the level of phosphorylated ATF2 in response to different patterns of JNK/p38 pathway, we created a rule-based mechanistic model (Fig. 7a and Supplementary Table 2). The MAPKs bound to their respective TAD regions, their activated form phosphorylated the T69/T71 phosphoswitch, and pp-JNK also phosphorylated S90. Binding affinities and phosphorylation rates were determined in vitro (Supplementary Fig. 7). Proteins are dynamically phosphorylated by protein kinases and dephosphorylated by phosphatases in the cell, and this reversible modification plays a key role in biological regulation. Therefore, the model was extended to include the action of upstream kinases and deactivating phosphatases for MAPKs, as well as phosphatases that counteracted MAPK mediated phosphorylation of ATF2. In order to focus on MAPK → substrate phosphorylation and keep the number of parameters low, we used a simplified model to implement MAPK activation and dephosphorylation where MAPKs had only two states: an unphosphorylated/inactive and a phosphorylated/active. MAPKs were activated by upstream kinases (k6 or k7) and deactivated by phosphatases (dp1 and dp2) at a single site, and the action of these enzymes on MAPKs were simply modeled as first order reactions. These enzymes are known to bind to the same surface on MAPKs as the substrates (e.g., ATF2), therefore, their binding is mutually exclusive[29]. ATF2 phosphorylation was counteracted by independent phosphatases acted on T69/T71 or S90 phosphorylation sites (dp3 and dp4). Parameters were determined based on the response of HEK293T cells to anisomycin treatment (Fig. 7b). Anisomycin stimulation experiments were carried out with wild-type, S90N, and MUT4 ATF2 constructs in the absence or presence of a JNK-specific inhibitor and p38-ATF2 binding

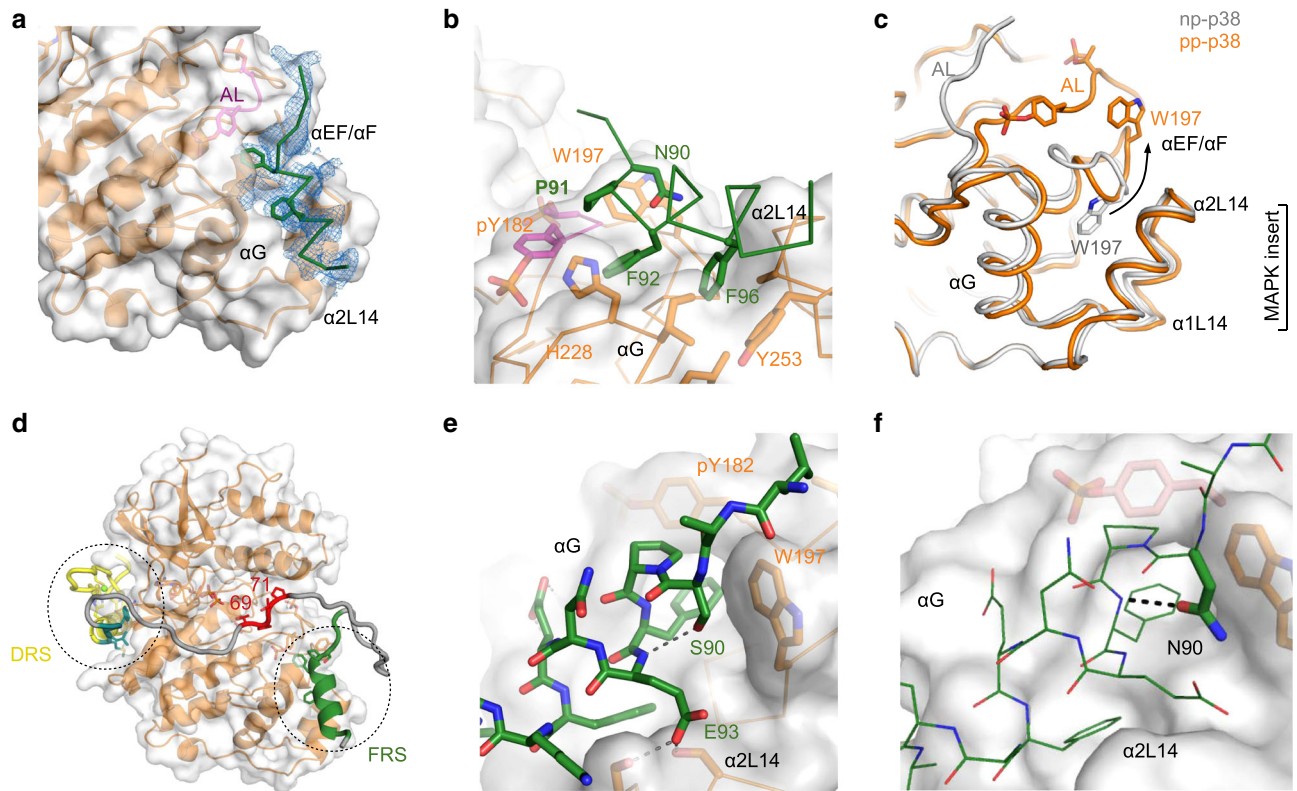

**Fig. 5 Structure of the pp-p38:ATF2(SPFENEF) complex. a** Crystal structure of the pp-p38:ATF2(83–102)_S90N complex. Residues between 88 and 101 are visible in the crystal structure as shown in the simulated annealing Fo-Fc omit map contoured at 2σ. **b** The helical ATF2 peptide is wedged in-between the N-terminal parts of the αG and α2L14 helices. The F-motif recruitment site (FRS) pocket (accommodating two phenylalanines, F92 and F96) is lined by activation loop residues (e.g., pY182) and W197. **c** Nonphosphorylated (np-p38) and phosphorylated p38 (pp-p38) crystal structures show a major difference in the orientation of the αEF/αF loop with W197. In the nonphosphorylated protein the FRS pocket is blocked by W197, while in the phosphorylated protein it forms the upper rim of this pocket together with the phosphorylated activation loop (AL). **d** Structural model of the pp-p38:WT-ATF2(TAD)complex. The model was generated using ambiguous restraints derived from NMR titration data on pp-38 and SPFENEF motif peptide. Zinc-finger + D-motif (yellow + cyan) and substrate peptide (T69 as the target site, red) were docked using unambiguous distance restraints (DRS: D-motif recruitment site). **e** S90 caps the ATF2 helix and is located at a critical position: phosphorylation at this site sterically blocks the binding interface and affects the conformation of the unbound peptide (Supplementary Fig. 6b). In addition, the two glutamate side-chains (E93 and E95) cap α2L14 and αG, and these hydrogen bonds are shown with dashed lines. **f** Structural basis of increased, non-regulatable binding for the invertebrate NPFENEF motif. An asparagine (N) at 90 is structurally more optimal compared to serine.

(i.e. NanoBit signal in live cells), MAPK and ATF2 probe phosphorylation (by Western-blots) were concomitantly monitored in time up to 40 min.

The mechanistic model was then used to calculate the in-cell phosphorylation level of the ATF2 T69/T71 phosphoswitch (pp-ATF2) under varying amounts of activated p38 and JNK levels (Fig. 7c). For ATF2-S90N, the JNK regulated S90 is OFF as this site cannot be modified and MAPK effects are additive towards building up high levels of pp-ATF2, and notably the inflection point is ~5% of the total flux for both MAPKs (see the left panels on Fig. 7c). In contrast, the 2D response surface is more complex with the ATF2 WT protein (where the JNK regulated S90 is ON and may get phosphorylated): (i) pp-ATF2 level stays low in response to low-to-medium p38 flux even under small amounts of JNK pathway flux, which increases the 2D dynamic range (i.e., the inflection point is ~25% of the maximum; see right panels on Fig. 7c), (ii) in general pp-ATF2 levels are more sensitive to JNK compared to p38 pathway flux, and (iii) increasing JNK flux under high pp-p38 levels can bring about maximum pp-ATF2 output only after going through a local minimum or threshold (see Fig. 7c, right panel, 3D plot).

The in silico model on MAPK controlled ATF2 phosphorylation was based on experimentally determined parameters

regarding MAPK and ATF2 TAD binding. Next, we were curious about how these binding parameters affect ATF2 phosphorylation. The parameters for JNK(DRS)-, p38(FRS)-, and p38(DRS)-ATF2 TAD binding were arbitrarily changed to simulate three distinct scenarios: (i) JNK would bind to Zn-finger + D-motif module 100-fold stronger (which could be achieved by increasing the $k_{on}$ of the interaction); (ii) p38 would bind to the SPFENEF motif 100-fold stronger, and (iii) p38 would bind with 10-fold stronger—both to the Zn-finger + D-motif and the SPFENEF motif—and JNK with 10-fold weaker affinity (Fig. 7d). In the first and second case, pp-ATF2 levels were less responsive or oversensitive to p38 activation, respectively, and in the third case, pp-ATF2 was less responsive to JNK activation.

In conclusion, the model showed that the MAPK binding affinity and the specificity of ATF2 TAD regions are set in vertebrates so that to allow graded phosphoswitch activation under broad p38/JNK pathway activity in the cell.

## Discussion

We found that an intact Zn-finger partially overlapping with the D-motif is the minimal JNK binding domain of ATF2. Interestingly, similar synergism between a small structured domain (PB1)

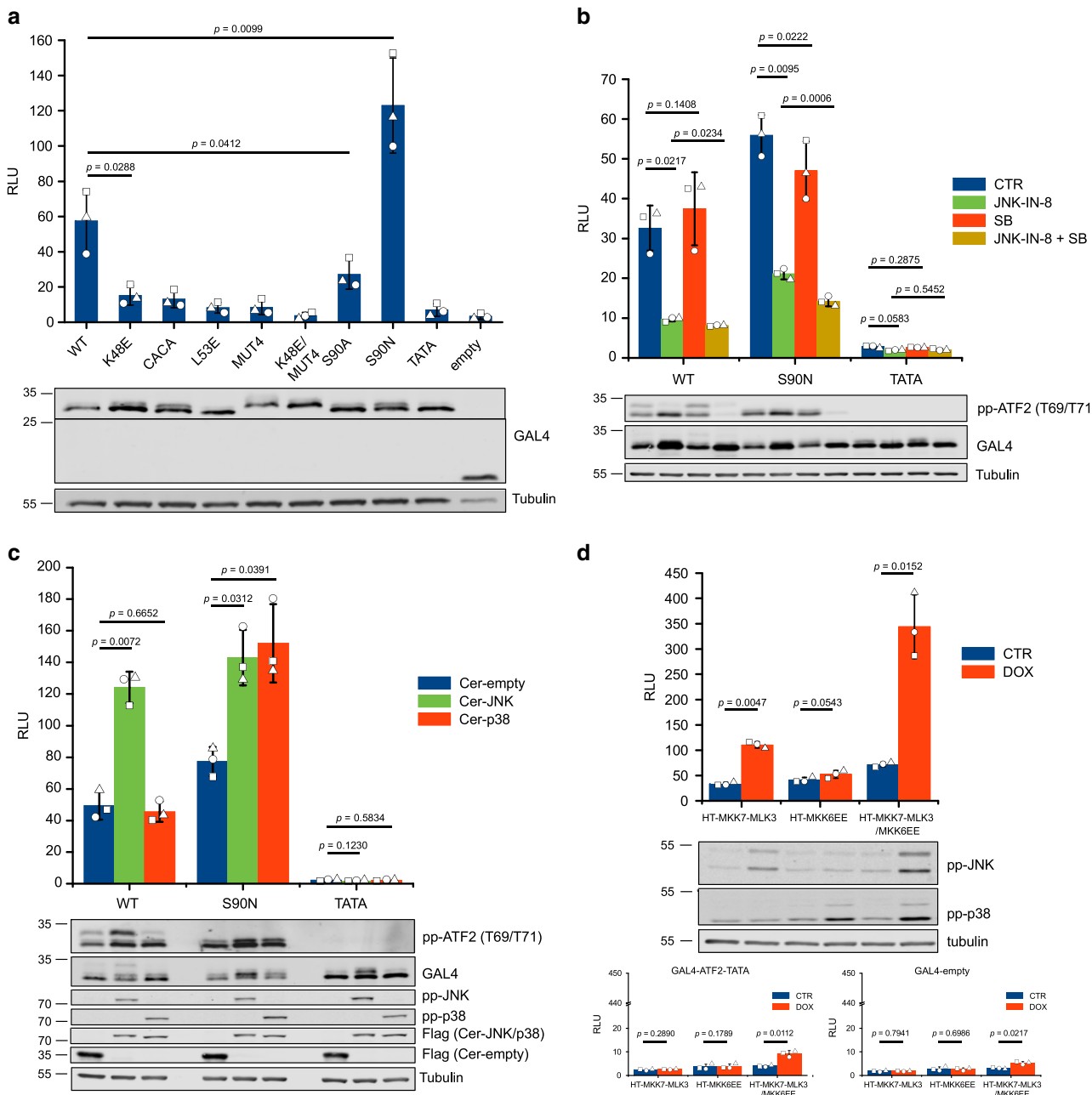

**Fig. 6 Co-regulation of ATF2 TAD mediated transcription by JNK and p38. a** The effect of different TAD residues on basal transcription. HEK293T cells were transfected with a GAL4-luciferase reporter plasmid and GAL4-TAD fusion expression constructs. Non-phosphorylatable GAL4-ATF2-TATA(T69A/T71A) and empty GAL4 constructs were also included as negative controls. RLU: relative luminescence unit (RLU). Panels below show results of western-blots: similar expression of different GAL4-TAD fusions were confirmed by using anti-GAL4 antibody and anti-tubulin antibody was used as load control. **b** The effect of JNK- or p38-specific inhibitors on TAD mediated basal transcription. Relative luminescence (RLU) was measured after treating cells with 1 μM JNK inhibitor (JNK-IN-8), 1 μM p38 inhibitor (SB, SB202190) or both inhibitors (JNK-IN-8 + SB), while 0.1% DMSO solvent was used as control (CTR). Panels below show western-blot results using anti-pp-ATF2(T69/T71), anti-GAL4 or anti-tubulin antibody. Note that the upper band on the pp-ATF2(T69/71) blot corresponds to GAL4-TAD fusions phosphorylated on S90. **c** Ser90 phosphorylation by JNK limits p38-mediated transcriptional activation. GAL4-ATF2-(19–100)-WT, -S90N, and -TATA constructs were co-transfected with Cerulean-empty-FLAG (Cer-empty), Cerulean-JNK1-FLAG (Cer-JNK) or Cerulean-p38α-FLAG (Cer-p38) expression plasmids. Phosphorylation of ATF2-TADs were detected by western-blot using anti-pp-ATF2(T69/T71) antibody. Expression of Cerulean constructs was detected using anti-FLAG antibody and increased levels of MAP kinase activation were confirmed by using anti-pp-JNK (T183/Y185) and anti-pp-p38 (T180/Y182) antibodies. **d** JNK and p38 synergistically increase TAD mediated activation. HEK293T Tet-on (HT) cell lines for inducible expression of MKK7-MLK3, MKK6-EE or both (MKK7-MLK3/MKK6-EE) were transfected with GAL4-ATF2-TAD(19–100)-WT (upper panel) or -TATA and empty constructs (lower panels). Endogenous MAPKs (JNK, p38 or both) were switched on by adding doxycycline (DOX). CTR—control, no DOX treatment. Error bars on all bar charts show SD calculated based on three independent experiments ($n = 3$). Bar charts show all data points and $p$-values calculated using a two-sided paired $t$-test. Source data are provided as a Source data file.

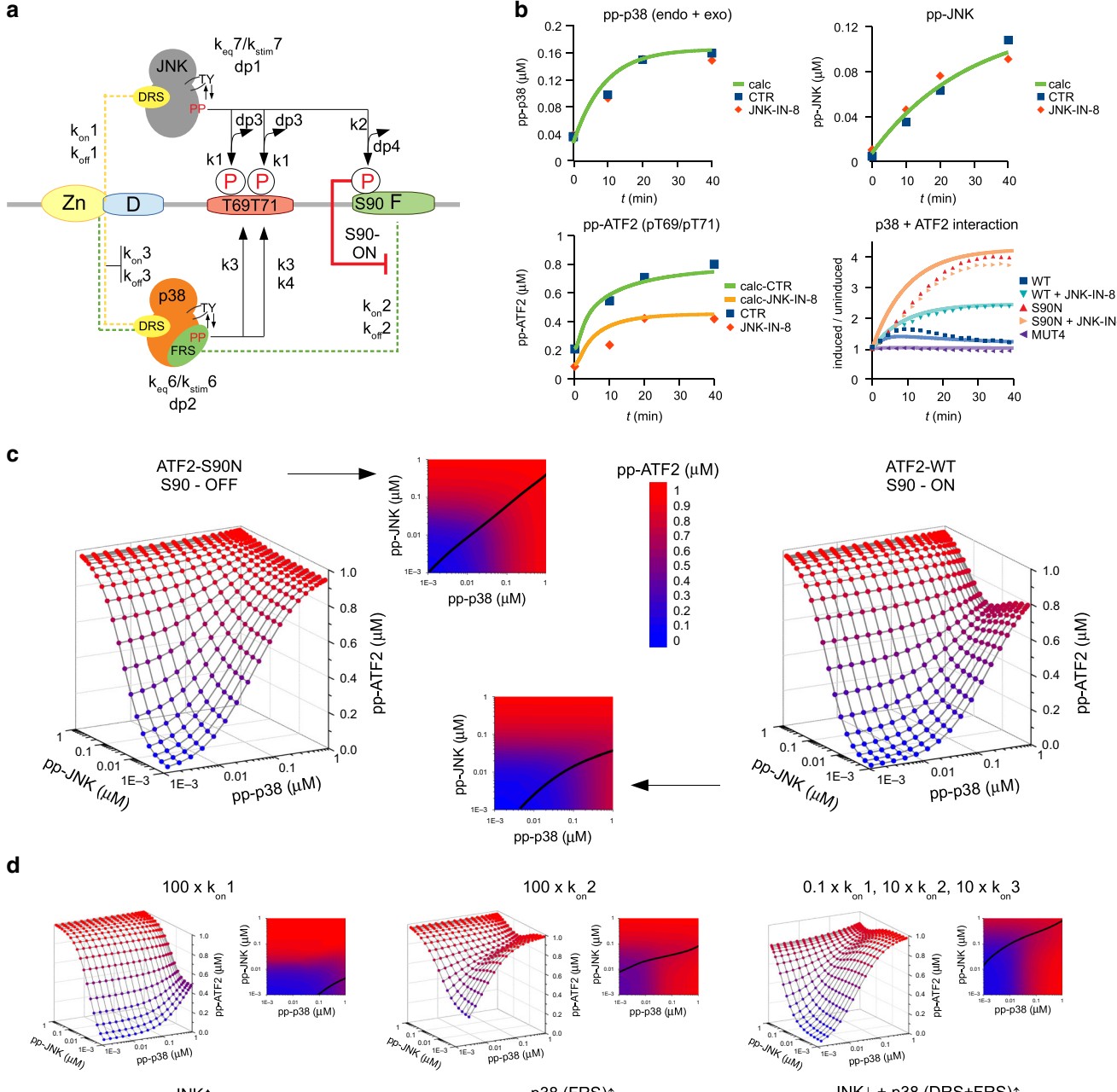

**Fig. 7 Modeling the impact of JNK and/or p38 MAPK pathway flux on ATF2 TAD phosphorylation. a** Mechanistic scheme of MAPK mediated ATF2 TAD phosphorylation. The panel shows the architecture of the rule-based BioNetGen model. Dashed lines represent binding rules (with $k_{on}$ and $k_{off}$) and lines with an arrow indicate enzymatic reactions (with reaction rates of kinases, k, and phosphatases, dp). Signaling is initiated by increasing k6 and k7 activity ($k_{eq}6 \rightarrow k_{stim}6$, $k_{eq}7 \rightarrow k_{stim}7$). D: D-motif; DRS: D-motif recruitment site; F: SPFENEF or F-motif; FRS: F-motif recruitment site; S90-ON: S90 is phosphorylated by JNK blocking p38(FRS):F-motif binding. P: phosphorylation on T69, T71 or S90. **b** Determination of model parameters. HEK293T cells were stimulated by anisomycin and pp-JNK, pp-p38 and pp-ATF2-T69/T71 were monitored in western-blots (JNK-IN-8: in the presence of JNK inhibitor; CTR: control, without the inhibitor; "calc" denotes simulated plots). The parallel p38-ATF2 NanoBit assay signal was also used to determine the parameters that could not be directly measured (i.e., $k_{stim}6$, $k_{stim}7$, dp1, dp2, dp3 and dp4). **c** Simulated ATF2 TAD phosphorylation levels 40 min after stimulation. The model was stimulated by different $k_{stim}6$ and $k_{stim}7$ (evenly distributed from $10^{-6}$ to $10^{-2}$ in the logarithmic scale). 3D plots show ATF2 TAD phosphorylation as the function of different amounts of pp-p38 and pp-JNK (from 0.001 to 1 µM; from blue to red, respectively), where pp-JNK and pp-p38 levels denote the activated state of the MAPKs. (Total in-cell MAPK and ATF2 concentrations were set to 1 µM for each protein.) ATF2-S90N S90-OFF: ATF2 TAD with a non-phosphorylatable serine to asparagine replacement at 90 (invertebrates); ATF2-WT S90-ON: results of the simulation with intact vertebrate ATF2 TAD. 2D panels show projections of the corresponding 3D plots. Black line indicates where pp-ATF2 level is equally sensitive to the activation of both MAPKs. The region above this line corresponds to an area where pp-ATF2 is more sensitive to JNK, while the region below the line is more sensitive to changes in p38 pathway flux. **d** The impact of MAPK binding affinity and specificity on pp-ATF2 levels. JNK-ATF2 or p38-ATF2 binding affinities were arbitrarily changed in the model and the impact of these on pp-ATF2 level was simulated. Binding affinity was increased by 100-fold by increasing the $k_{on}$ (left and middle panels) or changed by 10-fold (lowered for JNK but increased for p38; right panel). Note that $k_{off}1/k_{on}1$, $k_{off}2/k_{on}2$, and $k_{off}3/k_{on}3$ describe the affinity of JNK(DRS)-, p38(FRS)-, and p38(DRS)-ATF2 TAD binding, respectively (FRS: F-motif recruitment site; DRS: D-motif recruitment site).

and the MKK5 D-motif was described in the unrelated MKK5-ERK5 complex[30]. In addition to making extra contacts, the Zn-finger in the ATF2 TAD contributes to the binding energy by decreasing the entropic cost of binding in the MAPK D-groove. The ATF2 D-motif confers to a classical NFAT4-type MAPK binding consensus, and a linear D-motif would be sufficient to promote the phosphorylation of the T69/71 phosphoswitch[31]. However, we had showed earlier that D-motifs have only limited MAPK binding specificity[16]. Interestingly, the ATF2_CACA construct (in which the Zn-finger structure is disrupted) bound p38 better compared to wild-type while JNK binding decreased (Supplementary Fig. 5c). This suggests that the Zn-finger limits the capacity of the partially overlapping D-motif to bind to p38 and favors JNK binding. The Zn-finger+D-motif module is more JNK-specific compared to an NFAT4-type D-motif, and specificity is achieved through both positive and negative factors that promote JNK- and limit p38-binding, respectively.

Double-phosphorylated p38 bound ATF2 stronger than non-phoshorylated p38. This is similar to how ERK2 binds F-site peptides (e.g., the FxFP motif from ELK1)[32]. Although ATF2 does not have an FxFP motif, the SPFENEF motif found to be critical for pp-p38 binding also has two phenylalanine residues. Therefore, we posited that the SPFENEF region binds to the FRS, which was then confirmed by NMR based interface mapping and X-ray crystallography. How the residue at position 90 in the sequence affects pp-p38 binding is intriguing. An asparagine at this position mediates stronger binding (by about 5-fold) compared to a serine, while phosphorylation at Ser90 abolishes binding. The former may be explained by a more optimal side-chain at a critical position of the protein–protein binding interface. Conversely, a far less favorable chemical group (i.e. a highly charged and bulky phosphate) in the same position explains diminished binding.

The SPFENEF motif specifically binds to p38 and the structural basis of FRS binding specificity can be explained by the crystal structure of the pp-p38:ATF2(83–102) complex. The MAPK FRS pockets are overall similar but residues corresponding to W197 of p38 are different in ERK2 (serine) and JNK1 (methionine), thus ERK2 or JNK1 cannot form the FRS pocket required for ATF2 SPFENEF motif binding.

The JNK binding Zn-finger + D-motif module and FENEF motif-like regions are both present in most extant organisms even in sponges (Supplementary Fig. 1d). However, the JNK-binding module is absent while the p38-binding linear motif is clearly present in Drosophila (Supplementary Fig. 1e). In agreement with this, ATF2 is phosphorylated and regulated by p38 but not by JNK in this organism[33]. Furthermore, the fission yeast (Schizosaccharomyces pombe) ATF2 homolog (Atf1) is also known to be regulated by the mammalian p38 homolog (Spc1/Hog1)[34]. In conclusion, comparative sequence analysis of ATF2 homologs suggests that co-regulation of the ATF2 phosphoswitch by two distinct MAPKs is an ancient evolutionary trait, but the architecture of the ATF2 TAD have gone through important evolutionary changes regarding phosphoregulation.

According to the quantitative rheostat-like switch concept, the rate of transcription of a specific gene is continuously variable and coupled directly to the strength of intracellular signaling events[35]. Compared to the stable "on" and "off" modes of gene regulation, this model can be better applied to describe transcription initiation by immediate-early (IE) genes (e.g., AP-1 factors) that do not depend on new transcription/translation and are directly linked to signaling cascades. The primary response to an extracellular cue is triggered by TF TAD phosphorylation. In our study we found that the ATF2 TAD architecture is key in determining how the transcription factor will respond to JNK, p38 or to the concomitant activation of these two. Unexpectedly, ATF2 TAD does not simply integrate the signal coming from the

two MAPKs, but due to its unique architecture may sense the relative strength of these two MAPK pathways. This latter capacity of the ATF2 TAD goes beyond additive phosphorylation of the same phosphoswitch. We postulate that the basic integration capacity, emerging from an ancient pre-vertebrate ATF2 architecture comprised of JNK and p38 binding sites, became more complex due to an amino-acid change (Asn to Ser at 90) in the critical p38 binding region in vertebrates: one of the MAPKs (JNK) acquired the capacity to have a direct influence on how the other (p38) effects ATF2 mediated transcription. Unfortunately, the biological significance of this more complex regulation is not known yet. ATF2 is a constitutively expressed protein and plays a major role in controlling the transcription of other IE transcription factors and its knock-out in mice causes defects in skeletal and CNS development[36]. Tentatively, JNK/p38 based co-regulation of ATF2 activity may contribute to primary IE gene response diversity that is characteristic to different cell-types or development processes[35].

We have a substantial understanding on how individual MAPKs bind to TFs, since most of the known D- or F-motifs were originally found in these proteins decades ago. Physiological responses to extracellular cues depend on a complex activation pattern of different MAPKs (e.g. ERK1/2, JNK and p38), which is not exclusive but combinatorial, and not binary (i.e., on or off) but quantitative. Moreover, deviation from a normal pattern—which may depend on the cell type—leads to signaling related diseases such as cancer or inflammation[20,37], and now we know that even to drug resistance[38]. For ATF2 we showed how this MAPK substrate reads out different MAPK activation profiles. The mechanistic basis behind this seemingly complex process is simple: MAPK binding regions and their phosphorylation target sites are organized into a specific linear pattern. Ultimately, it is the TAD architecture around the transcription controlling phosphoswitch that is fine-tuned according to the signaling logic of the organism, but the building blocks—protein kinases, their binding and phosphorylation target motifs—are universal.

## Methods

**Protein expression and purification.** All primers and oligonucleotides used in this study are listed in Supplementary Table 3. All protein constructs are based on the human sequences. Full-length ATF2 (1–505; Uniprot: P15336-1) was amplified from HEK293T cDNA and then various fragments of the TAD were cloned into a modified pET-15 vector (Novagen) including N-terminal GST/MBP tag followed by a tobacco etch virus (TEV) cleavage site and a C-terminal hexa-histidine tag (Supplementary Table 3). For structural studies ATF2 TAD fragments were inserted into a modified pBH4 vector. Proteins were expressed with an N-terminal hexa-histidine tag that was cleaved off by the TEV protease. Protein constructs had a tyrosine inserted at the ATF2 N-terminus to enable protein concentration determination by absorbance at 280 nm. Mutant constructs were generated by two-step PCR mutagenesis. DNA expression plasmids were transformed into BL21(DE3) cells and grown in LB supplemented with 200 μM Zn-acetate until $OD_{600} = 0.5$, protein expression was induced with 200 μM isopropyl β-d-1-thiogalactopyranoside (IPTG) overnight at 18 °C. Cell pellets were lysed and the recombinant protein was purified using Ni-NTA agarose beads. Some ATF2 constructs were expressed as maltose-binding protein (MBP) fusions. A modified pET-15 vector (pET-MBP) was used to express MBP-ATF2-His6 constructs that were expressed similarly as above and purified on maltose resin after Ni-NTA purification. Protein samples were stored in MBP elution buffer (20 mM Tris pH 8, 150 mM NaCl, 10% glycerol, 2 mM TCEP, 0.1% BOG (Octyl-beta-Glucoside), 2 mM BA (benzamidine), 30 mM maltose) for fluorescence polarization (FP) based measurements or for kinase assays. For ITC measurements the MBP tag was removed by TEV protease and ATF2-His6 was further purified on a Resource S column in buffer containing 20 mM MES (pH 6.3), 25 mM NaCl, 10% glycerol and 2 mM βME and eluted with NaCl gradient. For crystallography His6-ATF2-(19–58)-WT or the Q34R/H47R mutant was captured on Ni-NTA agarose beads, the hexa-histidine tag was cleaved off by the TEV protease, the protein was purified on a Resource S column (pH 6.3) and gel-filtrated into storage buffer (20 mM Tris, pH 8, 150 mM NaCl, 2 mM TCEP) using a Superdex75 10/30 size-exclusion column (GE Healthcare). For NMR studies His6-ATF2-(19-78) or -(19-106) were transformed into BL21(DE3) and the culture (2 L) was grown in LB at 37 °C until $OD_{600} = 0.6$–0.8. Cells were pelleted and transferred to 0.5 L M9 media containing 1 g $^{15}NH_4Cl$, 200 μM Zn-acetate, 1× vitamin mix (Sigma,

#B6891), 1× MOPS micronutrient mix and 2 g glucose, then induced with 1 mM ITPG at 37 °C for 4 h. Labeled protein was captured on Ni-NTA resin; after cleaving the hexa-histidine tag by TEV protease, [15]N-ATF2 was further purified on Resorce S (19–78, pH 6.3) or Q column (19–106, pH 9) and gel-filtrated into storage buffer (20 mM Tris, pH 8, 150 mM NaCl, 2 mM TCEP).

JNK1-dc20 (where the C-terminal 20 residues are truncated), p38α and ERK2 were expressed in *E. coli* using standard procedures. Phosphorylated MAPKs were produced by co-expressing them with constitutively active GST-tagged MAP2Ks[16]. The purity of the protein samples were analyzed in SDS-PAGE. All of the protein samples were flash-frozen in liquid nitrogen and stored at −80 °C in buffers containing 10% glycerol. None of the purification, storage and assay buffers contained EDTA nor DTT so that to maintain the structural integrity of the ATF2 zinc-finger domain, but samples contained 2 mM TCEP.

SPFENEF peptide (ATF2: 83–102), its S90N and phospho-S90 versions were all chemically synthesized on an automated PSE Peptide Synthesizer (Protein Technologies, Tucson) with Fmoc strategy.

**Protein–protein binding assays.** GST pull-down experiments with GST-ATF2 constructs and GST control as baits and JNK1 as prey were performed using purified proteins in GST-binding buffer (20 mM TRIS, pH 8.0, 100 mM NaCl, 0.1% IGEPAL, 2 mM TCEP)[16]. Fluorescence polarization (FP) based protein–protein binding experiments were carried out using 50 nM carboxyfluorescein(CF)-labeled CF-evJIP (CF-PPRRPKRPTSLDLPSTPSL) for JNK1, and the data was analyzed using OriginPro. For direct measurements the concentration of JNK1 was increased in a titration experiment, then the binding of the unlabeled peptide was determined by titrating the JNK1-labeled peptide complex sample—at a JNK1 concentration corresponding to ~60–80% complex formation—with increasing amounts of the unlabeled peptide[16]. All of the FP measurements were performed in 20 mM Tris (pH 8.0), 100 mM NaCl, 0.05% Brij35P, and 2 mM TCEP and measured using a Cytation[TM] 3 (BioTek Instruments) plate reader in 384-well plates. ITC measurements were performed on MicroCal iTC200 (GE Healthcare) at 20 °C. Injection parameters were identical for all experiments (1.5 μL injections every 180 s, 20 in total at a stirring speed of 750 rpm). Before the experiments protein samples were dialyzed overnight in ITC buffer (50 mM HEPES, pH 8.0, 150 mM NaCl, 10% glycerol, 0.5 mM TCEP) in the absence or presence of 2 mM EDTA and subsequently concentrated. MAPK concentration in the measuring cell was 100 μM.

**Kinase assays.** MBP-ATF2-His6 protein constructs were double affinity purified and used as MAPK substrates. For ATF2 substrates with c-Jun and NFAT4 docking motifs, the ATF2 Zn-finger+docking motif region was replaced by the following docking peptide sequences: SNPKILKQSMTLNLADP and LERPSRDHLYLPLEP, respectively), which were followed by the ATF2 59-100 region. Substrates were dotted on nitrocellulose membrane and the assay was performed by incubating the membranes with the MAPK in kinase buffer (50 mM HEPES, pH 7.4, 100 mM NaCl, 5% glycerol, 0.05% NP-40, 2 mM TCEP, 2 mM MgCl2) supplemented with 1 mM ATP. Kinase assays were performed in kinase buffer containing 2 μM substrate and 2 nM enzyme. Reactions were stopped by adding SDS-loading buffer containing 20 mM EDTA and samples were subjected to western-blots. Enzyme kinetics measurements on ATF2 phosphorylation were performed in solution using 5 nM enzyme, MBP-ATF2(19–100) substrate (100, 50, 25, 10, 5, 2.5, 1, and 0.5 μM) and the reaction was started by adding 5 mM ATP. Competition assays with synthetic SPFENEF peptides were performed using 2 nM pp-p38, 2 μM MBP-ATF2 substrates and 3-fold dilution series of peptides starting from 1 mM. Reactions were stopped by adding kinase buffer containing 50 mM EDTA and diluted to 0.05 μM MBP-ATF2 in the same buffer. 1 μl of the diluted reaction mixture was spotted and dried on a nitrocellulose membrane in the presence of a calibration series. Calibration series were made from 2× dilution of fully phosphorylated MBP-ATF2 mixed with dephosphorylated MBP-ATF2 (0.05 μM in total) to confirm the linear range of the detection, and the rate of product formation was calculated. Membranes were developed using anti-phospho-T71 ATF2 antibody (Cell Signaling Technology, #9221) at 1:3000 dilution or anti-phospho-T69/T71 ATF2 (Merck, #05-891) at 1:5000 dilution or anti-HIS (Sigma, #H1029) at 1:10,000 dilution and a secondary anti-rabbit antibody (IRDye® 800CW Goat anti-Rabbit IgG, #926-32211) or secondary anti-mouse antibody (IRDye® 800CW Goat anti-Mouse IgG, #926-32210), respectively, at 1:10,000 dilution with LI-COR Odyssey® Clx infrared imaging system and Image Studio Lite. For $K_i$ measurements, the maximal velocity was determined from a reaction containing no peptide, and $IC_{50}$ values were obtained by using the logistic dose–response equation in QtiPlot. $K_i$ was calculated as $K_i = IC_{50}/(1 + S/K_M)$ where the $K_M$ was 17.3 μM (pp-p38–MBP-ATF2 (19–100)-WT).

**NMR spectroscopy.** For the JNK:ATF2 complex the [1]H-[15]N heteronuclear single-quantum coherence (HSQC) spectra were obtained using 50 μM uniformly [15]N-labeled ATF2-(19–78) in the absence or presence of JNK1 at molar ratios of 0.5:1 and 1:1 [JNK1]:[ATF2]. Experiments were carried out in 20 mM potassium phosphate (pH 6.3), 200 mM NaCl, 5% d-Glycerol and 2 mM TCEP containing 10% (v/v) D2O at 25 °C on an AVANCE II Bruker spectrometer (800 MHz) equipped with a TCI-active HCN cooled z-gradient cryoprobe. Published [1]H, [15]N

resonance assignment of ATF2-(19–106) were used to analyze the data[11]. Spectra were processed using NMRPipe[39] and NMR data were analyzed using NMRFAM-SPARKY[40]. Peak heights were used to calculate intensity ratios of ATF2 in the presence versus in the absence of JNK1.

To map out the p38 binding region of the ATF2 TAD experiments were carried out on a four-channel 600-MHz Varian NMR SystemTM spectrometer equipped with a 5-mm indirect detection triple resonance ([1]H[13]C[15]N) z-axis gradient probe. [1]H-[15]N HSQC spectra were performed on 100 μM uniformly [15]N-enriched ATF2-(19–106) in the absence and presence of pp-p38 and np-p38 at molar ratios of 0.5:1 and 1:1 ([pp-p38]:[ATF2]) and 2:1 ([np-p38]:[ATF2]). Experiments were carried out in 20 mM potassium phosphate (pH 6.3), 200 mM NaCl, 2 mM TCEP at 27 °C. For the analysis of the data, previously published [1]H, [15]N resonance assignment of ATF2-(19–106) was used[11]. Spectral processing and data analysis were carried out using Felix 2004 (Accelrys, Inc.) and CCPNMR. To assess the combined ([1]HN, [15]N) chemical shift perturbations, values of $\Delta\delta$([1]HN,[15]N) were calculated using the equation of $\Delta\delta_{1HN,15N} = \sqrt{(\Delta\delta_{HN})2 + (w1*\Delta\delta_N)2]}$, where w1 (= 0.154) is a weight factor determined using the BioMagResBank chemical shift database[41].

Production of np-p38 and pp-p38 for NMR measurements was done similarly as described before[28]. Briefly, expression of uniformly [[2]H,[15]N]-labeled p38 was achieved by growing cells in D2O based M9 minimal media containing 1 g/L [15]NH4Cl, respectively, and D-glucose as the sole nitrogen and carbon sources. Multiple rounds (0, 30, 50, 70, and 100%) of D2O adaptation was necessary for high-yield expression. The interaction between nonphosphorylated or phosphorylated p38 and the SPFENEF motif was studied by mixing the chemically synthesized peptide with 0.1 mM [[2]H,[15]N]-labeled np-/pp-p38 and the 2D [[1]H,[15]N] TROSY spectra of free and peptide added samples were compared. Samples were in 10 mM HEPES pH 7.4, 0.15 M NaCl, 5 mM DTT and 90% H2O/10% D2O. The spectra were processed using Topspin 4.0.6 and analyzed using NMRFAM-SPARKY. The NMR spectra were acquired at 35 °C on a Bruker Avance NEO 600 MHz NMR spectrometer equipped with a TCI-active HCN cooled z-gradient cryoprobe.

Resonance assignment of the SPFENEF peptides (ATF2 83–102) was obtained using the combination of 2D TOCSY, 2D NOESY and natural abundance [1]H-[13]C HSQC experiments. [1]H chemical shifts were referenced to external 4,4-dimetyl-4-silapentane-1-sulfonate (DSS) and the [13]C chemical shifts were referenced indirectly to DSS. Experiments were carried out on a four-channel 600-MHz Varian NMR SystemTM spectrometer equipped with a 5-mm indirect detection triple resonance ([1]H[13]C[15]N) z-axis gradient probe. Secondary structure propensities (SSP) of the peptides were calculated on the basis of $H_\alpha$, $C_\alpha$, and $C_\beta$ chemical shifts using the SSP program[42]. Briefly, the SSP score of a residue can be interpreted as the expected fraction of α-helical or β-strand/extended structure. In the lack of long-range distance restraints in NOESY spectra, dihedral angles derived from SSP scores were used as restraints in structure calculations with ARIA (Ambiguous Restraints for Iterative Assignment, version 2.2). Typically, in each of the seven iterations, the ten lowest energy structures were used as templates for the next iteration and the seven best structures were used for restraint violation analysis. The computational algorithm in the structure calculation employed torsional angle simulated annealing followed by torsional angle and then Cartesian molecular dynamics cooling stages. Structural refinement was completed in a water shell.

**HADDOCK modeling.** For the JNK-ATF2 complex, the input model ensemble for ATF2 was created manually based on the ATF2 TAD NMR structure (PDB ID: 1BHI) and D-motif containing JNK-peptide complexes in Coot (Supplementary Fig. 3b)[43]. The input model for JNK was the JNK1-pepNFAT4 crystal structure (PDB ID: 2XS0) after deleting the NFAT4 peptide. Docking was performed with the Haddock2.2 webserver with the Expert interface and unambiguous distance restraints were derived from MAPK:D-motif crystal structures (Supplementary Fig. 3c)[23]. For the first step of rigid-body energy minimization, 1000 structures were generated. 200 structures with the lowest energy from rigid-body docking were subjected to semi-flexible simulated annealing in torsion angle space followed by refinement in explicit water.

For the pp-p38:ATF2-WT(TAD) complex, the pp-p38 and ATF2 SPFENEF (83–102) structure was extracted from the crystal structure of the pp-p38-ATF2 (S90N) complex, and N90 were mutated to serine in Coot. The ATF2 Zn-finger + D-motif was taken from the JNK-ATF2(Zn-finger + D-motif) crystal structure and the substrate motif (67-DQTPTP-72) was modeled as in the DYRK1A-substrate peptide complex (PDB ID: 2WO6)[44]. Unambiguous distance restraint were used for docking the Zn-finger + D-motif to pp-p38 similarly as described for the JNK-ATF2 complex (see Supplementary Fig. 3c). For the docking of the substrate motif (DQTPTP) to pp-p38 at the active site, unambiguous distance restraints were derived from the DYRK1A-peptide complex, where the T69 was treated as the phosphorylation target site. SPFENEF docking was driven by intermolecular (ambiguous) restraints based on perturbations on the pp-p38 2D [[1]H,[15]N] TROSY spectra. Active residues in the FRS site of p38 were chosen based on the results of the pp-p38:SPFENEF peptide (1:4) sample. Residues with >50% intensity decrease and with >10% solvent accessibility (RSA) were defined as active. Passive residues were defined as solvent exposed (RSA values of 10% or above) and contiguous to active residues. Active residues were limited to the MAPK FRS (or also called DEF

pocket), since the pp-p38-ATF2(S90N) crystal structure indicated this MAPK region to be the major contact surface, moreover, mutating this region in p38 had been shown to negatively affect p38-mediated ATF2 phosphorylation[45,46]. In the SPFENEF peptide F92 and F96 were treated as active residues, and its neighbors as passive residues. 1000 docked models were created through a rigid body minimization protocol, then the top scoring 200 structures underwent a semi-flexible simulated annealing protocol. Finally, all the resulting structures underwent refinement in explicit solvent. The three ATF2 peptides were stitched together in MODELLER[47].

**Protein crystallization and X-ray structure solution.** JNK1-dC20 and ATF2-Q34R/H47R(19–58) were mixed at 1:1.5 ratio (240 μM JNK1 + 360 μM ATF2) in the presence of 5 mM TCEP and 1 mM adenosine-5'-[(β,γ)-imido]triphosphate (AMP-PNP). Crystals grew in 8% w/v PEG 20 000, 10% v/v PEG MME 550, 0.025 M MgCl2, 0.025 M CaCl2, 0.08 M bicine/Trizma base (pH = 8.8) in hanging drops at 4 °C and the reservoir solution was 1.25 M NaCl. Crystals were flash cooled in liquid nitrogen after adding 2 μl crystallization solution with increased PEG concentration (11% w/v PEG 20,000 and 22% v/v PEG MME 550) to the drop. Data were collected at PETRA III beamlines, Hamburg, and processed by XDS and CCP4i. The structure was solved by molecular replacement using PHASER[48]. The JNK1 search model contained the full polypeptide chain of the MAPK from the JNK1-NFAT4 protein-peptide structure (PDB ID: 2XS0), while the ATF2 search model was from the solution structure of the zinc-finger domain (PDB ID: 1BHI, model 1). The MR solution contains two JNK-ATF2 complexes. The ATF2 C-terminal docking motif region, the ATF2 N-terminal residues, the JNK activation loop (chain B) and AMP-PNP were manually built in Coot. Structure refinement (2.7 Å resolution) was done using PHENIX with NCS restraints for corresponding polypeptides[49]. Structural figures were prepared using PyMol.

For the p38-ATF2 complex, pp-p38 and pepATF2_S90N(83–102) were mixed together at 1:5 ratio (250 μM pp-p38 + 1250 μM peptide) and supplemented with 5 mM TCEP and with an ATP competitive inhibitor (375 μM, Skepinone-L). Crystals grew in 14% PEG3350, 0.1 M cacodylate pH 6.5 in hanging drops at 4 °C and the reservoir solution was 1.375 M NaCl. Crystals were flash cooled in liquid nitrogen in the crystallization solution containing 30% glycerol. Data were collected at PETRA III beamlines, Hamburg. The structure was solved similarly as described for the JNK-ATF2 complex. The search model for molecular replacement was phosphorylated p38 from the crystal structure of the pp-p38:MK2 binary complex (PDB ID: 6TCA). The final crystallographic model contains one pp-p38:ATF2 (83–102):Skepinone-L complex where ATF2 residues 88-101 could be traced in the electron density map (1.95 Å resolution). Details of the data collection and structure refinement are found in Supplementary Table 1.

**Cell culture.** HEK293T cells (purchased from ATCC, CRL-3216) were cultured in Dulbecco's modified Eagle's medium (DMEM, Gibco) containing 10% fetal bovine serum (FBS) and 50 μg/ml gentamicin at 37 °C in an atmosphere of 5% CO2. For preparing the HT-M cell lines, HEK293T cells were transfected with pEBDTet vectors containing a MAP2K-MAP3K kinase domain fusion or constitutively activated versions of full-length MAP2Ks: MLK3-MKK7 and MKK6EE that turn on JNK and p38, respectively[26]. HEK293T Tet-on (HT) stable cell lines (HT-MLK3-MKK7 and HT-MKK6EE) were established by keeping the cells under puromycin or hygromycin for at least one week, then the expression of the FLAG-tagged transgene and concomitant specific MAPK activation were monitored by western-blots after doxycycline (DOX) treatment (2 μg/mL) in DMEM without FBS. Cells were transfected with Lipofectamine 3000 reagent and were kept under puromycin (5 μg/mL) or hygromycin (100 μg/mL) for selection.

For analyzing the phosphorylation of full-length ATF2 mutants by endogenous JNK, HT-MLK3-MKK7 cells were seeded in 24-well plate and were transfected the next day with 250 ng F1-ATF2 (1–505) constructs with Lipofectamine 3000 in Optimem. The medium was changed to DMEM 4 h after transfection and cells were induced for 0, 8 or 24 h with 2 μg/mL doxycycline. Cells were harvested in SDS loading buffer and subjected to SDS-PAGE, then samples were transferred to nitrocellulose or PVDF membrane. Anti-phospho-ATF2-T69/T71 (Merck, #05-891; 1:5000 dilution) and anti-phospho-p38 (Cell Signaling, #9215; 1:3000 dilution) signals were detected on nitrocellulose membrane, while anti-phospho-JNK (Cell Signaling, #9251; 1:2000 dilution), anti-FLAG (Sigma, #F1804; 1:10,000 dilution) and anti-tubulin (Sigma, #T6199; 1:10000 dilution) were detected on PVDF membrane using appropriate secondary antibodies (IRDye® 800CW Goat anti-Rabbit IgG, Li-Cor, #926-32211; IRDye® 800CW Goat anti-Mouse IgG, Li-Cor, #926-32210; or IRDye® 680RD Goat anti-Mouse IgG, Li-Cor, #926-68070; 1:10,000 dilution).

**Cell-based protein–protein interaction assays.** Yellow fluorescence protein (YFP) based BiFC assay was performed with YFP constructs split at position 159, and fragments (F1 and F2) were pasted into pcDNA 3.1 vectors (Invitrogen) to monitor interaction and localization of the JNK-ATF2 complex in HEK293T cells[50]. JNK1 was expressed as N-terminal F2 fusions and ATF2(1–505) as N-terminal F1 fusion. Full-length c-Jun (1–331, UniProt ID: P05412-1) was cloned into pcDNA 3.1 vector containing C-terminal FLAG-tag or N-terminal F1 fusion. For imaging, cells were seeded to 8-Well Glass Bottom μ-Slide plate

(#80827, Ibidi) and fixed with 4% paraformaldehyde solution 24 h after transfection, stained with 0.1 μg/ml DAPI and imaged with EVOS M7000 Imaging System (Thermo Fischer Scientific). Images were processed by the ImageJ software and nuclear localization of the fluorescent signal was calculated as a percentage of the total fluorescence intensity.

For luciferase complementation NanoBiT (Promega) assays[51], $2 \times 10^4$ cells per well were seeded onto a white 96-well plate (Greiner, 657160) 24 h prior to transfection. Transient transfections were carried out with Lipofectamine 3000 reagent according to the NanoBiT system's instructions. cDNAs were sub-cloned into Lbit and Sbit expression vectors: MAP kinases (JNK1 or p38α) were expressed as N-terminal Lbit fusions and ATF2 constructs as N-terminal Sbit fusion proteins. The FBS was removed 4 h after transfection and cells were starved for 20 h before assayed using the Nano-Glo reagent (Promega) according to the manufacturer's instructions in a Cytation 3 plate reader (BioTek). MAPK stimulation was initiated by adding 10 μg/mL anisomycin (#A9789, Sigma-Aldrich) at 37 °C and 5% CO2 level. Transfected and treated cells were subjected to western-blots using anti-phospho-ATF2-T69/T71 (Merck, #05-891; 1:2000 dilution), anti-phospho-c-Jun-S63 (Santa-Cruz, #sc-822; 1:2000 dilution) and other phospho-specific antibodies mentioned above. In the case of HTM-6 cell line, 2 μg/ml doxycycline was added 20 h prior the measurement to induce MKK6-EE expression and the luciferase activity was measured (for 10–15 min) after addition of 10 μM luciferase substrate (Coelenterazine h).

**Transcription activation assays.** ATF2-TAD (19–100) constructs were subcloned into pBIND vector as N-terminal fusion of GAL4 DNA-binding domain (1–147). The pBIND vector also expresses Renilla luciferase as a transfection control. $2 \times 10^4$ HEK293T cells per well were seeded onto a white 96-well plate 24 h prior to transfection and co-transfected with 50 ng GAL4-ATF2 and 50 ng 5×GAL4-Firefly-luciferase reporter plasmid (pGL4.31) using Lipofectamine 3000 in DMEM. 4 h after transfection the medium was changed to DMEM, and cells were serum starved for 20 h and assayed with Dual-Glo® Luciferase reagents (Promega, #E2920) according to manufacturers instruction. RLU (Relative Luminescence Unit) were determined as Firefly Luciferase activity/Renilla Luciferase activity. Inhibitor treatments were performed adding 1 μM JNK-IN-8 (Selleckchem, #S4901), 1 μM SB202190 (Sigma-Aldrich, #S7067) or 1 + 1 μM JNK-IN-8 + SB202190 8 h before measurement (final DMSO concentration was held at 0.1%). 10 ng Cerulean-empty-FLAG, Cerulean-JNK1-FLAG and Cerulean-p38α-FLAG were co-transfected with 50 + 50 ng GAL4-ATF2 and reporter vector where indicated. HT-MLK3-MKK7(Puro), HT-MKK6EE(Hygro) or HT-MLK3-MKK7/MKK6EE(Puro+Hygro) cell lines were used to activate endogenous MAPKs by adding 2 ug/mL doxycycline 16 h before the measurement. Samples were subjected to western-blot analysis and GAL4-ATF2 expression was detected using anti-GAL4 antibody (Santa-Cruz, #sc-510; 1:5000 dilution) or phospho-specific antibodies mentioned above. All experiments were performed independently at least 3 times and tested with paired Student's t-test, two-sided. Visualization and statistical analysis was done using LibreOffice Calc.

**Simulation of TAD phosphorylation.** Rule-based network modeling was carried out with the software package BioNetGen with the ordinary differential equation solver running on a desktop PC (Supplementary Software 1)[52]. Pathway activation was initiated from a pre-equilibrated state with baseline k7 ($k_{eq}$7) and k6 ($k_{eq}$6) activity (see Fig. 7a). To mimic stimulation by anisomycin, signaling was started by increasing k7 ($k_{stim}$7) and k6 ($k_{stim}$6) activity. The BioNetGen model includes the rules that JNK-ATF2 binding is not affected by the phosphorylation state of proteins and ATF2 binds to the DRS of JNK. ATF2 binds to the DRS and FRS of pp-p38 in a bipartite manner ($k_{on}$2, $k_{off}$2), but only when p38 is phosphorylated and ATF2 is not phosphorylated on S90. p38 binds to the DRS with only low affinity ($k_{on}$3, $k_{off}$3, $K_D$ ~1 mM) when bipartite binding is not possible. Kinetic parameters for ATF2-JNK ($k_{on}$1, $k_{off}$1) and ATF2-p38 ($k_{on}$2, $k_{off}$2) binding and rate constants for ATF2 T69 and T71 phosphorylation were determined based on in vitro kinase assay data (see Supplementary Fig. 7). Parameters were adjusted to the data using the differential evolution algorithm of PyBioNetFit[53]. The rate of T69 and T71 phosphorylation by JNK was set to be equal (k1). In the case of p38 mediated phosphorylation events, the T71 phosphorylation rate of T69 phosphorylated ATF2 was set to be slower (k4), compared to the normal p38-mediated T69/T71 phosphorylation rates (k3)[54]. The rate of S90 phosphorylation by JNK was derived from [γ-32P]ATP based in vitro kinase assay data using the ATF2-TATA substrate (k2) (see Supplementary Fig. 7c).

The results of in-cell NanoBit measurements with MAPK and ATF2 PPI probes and related Western-blot data were used to determine k7 or k6 mediated phosphorylation and phosphatase mediated dephosphorylation rates upon anisomycin treatment (see Fig. 4). The NanoBit assay data were simulated in a simplified system consisting of ATF2, JNK and p38 (1 μM), and additional exogenous p38 (1.65 μM). The level of activated JNK after 40 min of anisomycin treatment was set to be 10% of the total. Western-blot data of pp-JNK were used in PyBioNetFit to determine kinase ($k_{stim}$7) and phosphatase (dp1) rates acting on JNK. Basal activity of JNK was introduced with the $k_{eq}$7 rate constant. For p38 involving reactions, two sets of NanoBit data on the ATF2-p38 complex—CTR and JNK-IN-8 treated—were used for the fitting of $k_{stim}$6, dp2 and dp4 with PyBioNetFit. Basal activity of p38 was introduced with the $k_{eq}$6 rate constant.

These model parameters predicted that 5% of p38 gets activated after 40 min of anisomycin treatment. The phosphatase activity on T69/T71 (dp3) was determined using the pT69/pT71 Western-blot signal, assuming that 80% of ATF2 are phosphorylated after 40 min in the control experiment (CTR). In the case of JNK-IN-8 treatment, k1 and k2 were set to 0, and in the case of the S90N mutant, $k_{on}2$ was set 4.5x fold higher compared to WT, and k2 was set to 0 (see Supplementary Table 2). Parameter scanning with various $k_{stim}6$ and $k_{stim}7$ rates was performed using the final model and endogenous protein phosphorylation levels were calculated for 40 min after stimulation.

**Reporting summary**. Further information on research design is available in the Nature Research Reporting Summary linked to this article.

## Data availability
The data that support this study are available from the corresponding author upon reasonable request. The crystal structure of the JNK-ATF2(19–58) and the pp-p38α: ATF2(83–102) complex were deposited in the Protein Data Bank (PDB) with the accession code 6ZR5 and 6ZQS, respectively. The following X-ray structures are available from the PDB: 1BHI, 2XS0, 2WO6, 6TCA. Source data are provided with this paper.

## Code availability
Computer-readable model implementation written in BNGL is provided as Supplementary Software 1.

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

## Acknowledgements

The authors thank Marie Bogoyevitch for her support and valuable discussions. This work was supported by the National Research Development and Innovation Office (NKFIH) grants (NN 114309, KKP 126963 awarded to A.R.), the Hungarian of Academy of Sciences (KEP-10/2019), and by a National Institute of Health (NIH) grant 1R01GM100910 to W.P.

## Author contributions
K.K. designed and carried out the experiments, analyzed data and, wrote the paper. A.R. conceived the project, analyzed data, supervised research and wrote the paper. A.Z. mapped out the JNK binding region of ATF2. O.T. analyzed the structure of ATF2 TAD fragments, Ashish S./P.G. characterized JNK-ATF2 binding, and O.T./G.S.K./W.P. elucidated pp-p38-ATF2 binding by NMR spectroscopy. I.B. collected X-ray data. P.S., Anna S., P.E., Á.L.P., and A.A. contributed by carrying out some of the biochemical or cell-based experiments.

## Competing interests
The authors declare no competing interests.
