## [Peer Review File · Nature Communications]

REVIEWER COMMENTS

Reviewer #1 (Remarks to the Author):

This manuscript by Kirsch et al explores recognition the transcription factor ATF2 by the MAP kinases p38 and JNK. A major role of MAP kinases is to influence gene transcription through direct phosphorylation of transcription factors. It has been recognized for some time that MAP kinases target transcription factors through so-called “docking” interactions that occur separately from the sites of phosphorylation. ATF2, a common substrate of p38 and JNK, harbors a docking site (D-site) that binds to a conserved region present in both kinases. The authors report here that additional interactions are required for kinase-substrate binding, but surprisingly these interactions differ between JNK and p38. Details of both types of interaction are rigorously characterized by NMR, X-ray crystallography, and mutagenic analysis in vitro and in cells. The authors make two key insights into why the system would be set up this way. First is that based on the location of the interaction site, p38 only binds ATF2 when in its active state. Second, that there is Ser residue adjacent to the unique p38 interaction site that is phosphorylated by JNK, and phosphorylation at this site disrupts the ATF2-p38 interaction. The authors build a quantitative model showing how these phenomena influence the level of ATF2 phosphorylation in response to graded combinations of p38 and JNK activity. This is an interesting paper that greatly expands our understanding of how MAP kinases target transcription factors, and ways in which the mode of substrate recognition influences the systems level behavior of transcription factor activity.

The work is carefully done and I have no recommendations for additional experiments. However, as illustrated below there are aspects of the manuscript that could be presented in a clearer way.

Specific comments:

1. The description of the progression from the NMR data to the HADDOCK model to the ATF2-JNK X-ray crystal structure is a bit confusing and requires clarification. Figure 2B presumably shows the HADDOCK model that was used as the basis for mutagenesis experiments, but this is not stated. Or is it that the HADDOCK model was made to provide evidence that the double mutant used for crystallography does not impact the overall binding mode? The experiments shown in Fig 2B appear to be from competitive FP assays (since the associated data is in Fig S4) but this should be stated in the main figure legend.

2. Overall there is an excessive amount of supplemental data. The authors should consider moving some of this to the main text and removing extraneous material entirely. For example, Figure S3E

and the associated legend make two interesting points: that the prior crystal structure of JNK in complex with the ATF2 D-site peptide fragment incorrectly places Lys48 and His49, and also that in the JNK-NFAT4 D-site complex, the peptide backbone overlays well with a helical region of the ATF2 Zn finger. These are key points that should be made in the main text and not buried in the supplement. Another example: Figures S5A/B show deletion/mutagenesis mapping of ATF2 in p38 kinase assays that should be quantified and shown in the main text.

3. p. 13 1st paragraph, can the authors clarify and provide context for the statement “interestingly, the interaction is in slow exchange”.

4. The quantitative modeling of ATF2 is interesting, but I’m confused about the point being made in panel 7D showing the theoretical impact of perturbing MAPK-ATF2 interaction parameters. The perturbations are pretty extreme – 100x changes in affinities that are probably unrealistic. Also, can the authors comment on the relationship between p38/JNK activity on pathway dynamics or maximal levels of activation? I don’t get a sense of this from the way they present the model, but it must provide that information. Finally, could the authors speculate on what cellular conditions would provide differential regulation of p38 vs. JNK?

Reviewer #2 (Remarks to the Author):

Concerning Kirsch et al: "Co-regulation of the transcription controlling ATF2 phosphoswitch by JNK and p38"

There is a lot to like about this manuscript.

It is data-rich. Fig. 1 shows by deletion mutagenesis the sequences in ATF2 and cJun that mediate interactions with JNK1 integrating data from pulldowns and ITC experiments, and show that a Zn-finger participates. Fig/ 2 maps the interactions of ATF2 with JNK and p38. Key mutants are used to distinguish regions of ATF2 that participate in the interaction, including a tighter binding mutant that was used for crystallography. Fig. 3 presents the anisomycin induced phosphorylation of p38 and JNK showing the JNK uses the D-motif, and p38 uses the F-motif. Fig. 4 JNK phosphorylation of S90 reduces p38 interactions of ATF2. Fig 5 Structure! of p38-ATF2 (F-site binding). Fig. 6 Synergy between JNK and p38 in transcriptional activation.

In general, the data is clearly presented.

This paper builds nicely on prior knowledge, especially an NMR structure of ATF2 and a D-motif JNK complex structure.

They used a very good technique to activate JNK and p38 separately by expressing constitutively active forms of their cognate MEKs.

Best of all, they discovered that an evolutionary newcomer JNK phosphosite reduces p38 interactions with ATF2. Through structure they determined the mechanism of the p38 interaction and how it would be inhibited by S90 phosphorylation.

They have data supporting the kinetic modeling presented. However, as they point out in the discussion, the real function of the JNK inhibition of p38 ATF2 activation is far from clear, which diminishes the significance of the kinetic models.

Comments on the figures.

Fig. 1. Panel A sequence is too small to read. It needs to highlight all of the amino-acids under discussion including K48. It could also highlight the residues discussed in Fig. 2

Fig. 2. B has some lettering that is too small.

Comments on the writing.

1. In a couple of places, adding a phrase would make the paper much more accessible.

a. page 7 "We examined JNK1-ATF2 binding by NMR Spectroscopy." The phrase should be added "based on an existing NMR spectral assignment and structure for ATF2 and crystal structure of JNK-peptide interactions."

b. page 10 legend "Coelenterazine" should be replaced with "the luciferase substrate Coelenterazine."

2. The introduction. The introductory paragraph introduces concepts not known to non-experts, especially Mediator complex. The word "instrumental" implies action, "essential" would be better.

3. The conclusion. The conclusion is over-long. One paragraph that can go concerns prolactin hormone on p 22. The paragraph on Elk-1 (same page) should also go. All of these ideas (self-limiting responses, gene regulatory evolution) have been well discussed in many settings, so the purpose of bringing out these specific examples was lost on me. Best to highlight your first paragraph on p. 23 without these distractions.

--Elizabeth Goldsmith

Reviewer #3 (Remarks to the Author):

The authors explored how two different families of MAPKs, the JNK and p38 kinases, regulate the ATF2 transcription factor. First using different biochemical techniques, they found that a minimal region (19-58) that contains both the Zn-finger and the D-motif was necessary for the formation of the JNK1 and ATF2 complex, whereas neither of these motifs alone was insufficient for the binding. Using X-ray crystallography/NMR techniques, they established the structural and mechanistic basis of the differential ATF2 phosphorylation by JNK and p38. The authors developed a structural model of the JNK-ATF2 complex using ATF2(19-58) construct. Next, they compared p38 and JNK binding to the ATF2 transcription activating domain (TAD) and showed that it is mainly controlled by p38 when both kinases were activated. Moreover, the authors showed that JNK mediated phosphorylation decreased binding of activated p38 to ATF2 TAD. At the same time, further experiments showed that JNK and p38 synergistically increased TAD mediated activation. To better understand these experimental observations, the authors developed a rule-based, kinetic model. Although the phosphorylation and dephosphorylation of JNK and p38 was described by simple linear kinetics, the model fitted to its limited purpose, which is deriving the rules for the JNK and p38 binding and phosphorylation of ATF2 based on experimental observations. At the same time, the model has several weak points, and the statements that present this model as a model describing 'in-cell', i.e., the physiological situation, are incorrect (see comments). In summary, this paper contains interesting results and can be published after a major revision.

Comments:

1. "The in-cell parameters were determined based on the response of HEK293T cells to anisomycin treatment"

- The statement is incorrect. For full kinase activity, both T183 and Y185 of the activation loop of JNK must be phosphorylated. JNK is phosphorylated preferentially on Tyr by MKK4 and on Thr by MKK7. There are two main MAPKKs which activate p38, MKK3 and MKK6, and p38 is phosphorylated on T180 and Y182. The model accounts only for a single site (de)phosphorylation of JNK and p38 by MKK7 and MKK6, and the cognate phosphatases. In addition, JNK feedback phosphorylates and further activates MKK7 (doi: 10.1126/scisignal.aab0990). Because the authors do not know the kinetic reaction laws and have not measured the kinetic parameters of (de)phosphorylation, physiological, in-cell activation of JNK and p38 was not modeled.

2. "Therefore, the model was extended to include upstream kinases and deactivating phosphatases for MAPKs, as well as phosphatases that counteracted MAPK mediated phosphorylation of ATF2."

- The statement has to be tone down. A model does not explicitly consider kinases upstream JNK and p38. The phosphatases in the model are introduced as merely first order reactions.

3. "The mechanistic model was then used to calculate the in-cell phosphorylation level of the ATF2 T69/T71 phosphoswitch (pp-ATF2) under varying amounts of pp-p38 and pp-JNK levels. Double-phosphorylated p38 bound ATF2 stronger than non-phosphorylated p38"

- Again, only a single (de)phosphorylation site was modeled for p38 and JNK. Can the authors comment on that in the paper?

4. The model describes the binding of p38 to ATF2 that involves two ATF2 sites as the simultaneous binding to both sites. In addition, p38 can bind to ATF2 at only one site, DRS. However mechanistically, in terms of chemistry, the binding to two sites is likely sequential, rather than simultaneous. For instance, following the first binding to DRS, the subsequent binding to the FRS can occur. The rule-based model can readily include all these potential chemical scenarios.

5. In the model a JNK inhibitor does not affect JNK binding to ATF2, while decreasing the JNK catalytic phosphorylation constant to zero. Is this assumption supported by the experimental data?

Minor:

1. "D-motifs are normally located 10-50 amino acids N-terminal,"

- Suggested: D-motifs are normally located 10-50 amino acids (FROM THE) N-terminal, ...

2. "This data was used to create a mechanistic model on cellular TAD phosphorylation ..."

- Suggested: This data was used to create a mechanistic model OF cellular TAD phosphorylation ...

3. "The transactivating region of both cJUN or ATF2 contains a MAPK binding D-motif"

- Suggested: The transactivating region of both cJUN AND ATF2 contains a MAPK binding D-motif

4. "Next, we assessed the role of JNK binding on MAPK target site phosphorylation. The Znfinger+D-motif module (19-58) was left intact or replaced by cJUN or NFAT4 D-motifs and phosphorylation of the 69-TPTP-72 target site by activated JNK was monitored by anti-phospho immunoblots (Figure 1E). This analysis confirmed the importance of K48 and K35 in ATF2 and cJUN phosphorylation, respectively."

- Please clarify this conclusion, because a large Znfinger+D-motif module and not only K48 or K35 was substituted by other motifs.

RESPONSE TO REVIEWER COMMENTS

The reviewers' comments are in italic type and our responses are given in normal type.

Reviewer #1:

Specific comments:

1. The description of the progression from the NMR data to the HADDOCK model to the ATF2-JNK X-ray crystal structure is a bit confusing and requires clarification. Figure 2B presumably shows the HADDOCK model that was used as the basis for mutagenesis experiments, but this is not stated. Or is it that the HADDOCK model was made to provide evidence that the double mutant used for crystallography does not impact the overall binding mode? The experiments shown in Fig 2B appear to be from competitive FP assays (since the associated data is in Fig S4) but this should be stated in the main figure legend.

Yes, Figure 2B shows the HADDOCK model, which is now explicitly stated in the figure legend. We used this as a structural template to design additional interactions at the protein-protein interface. The WT ATF2:JNK complex did not give crystals and we were keen on increasing the binding affinity, as based on our experience MAPK:docking peptide complexes has a higher chance to crystallize as a complex if the binding affinity were submicromolar. Indeed, the Q34R/H47R ATF2 mutant - with about 100-fold stronger binding compared to WT – could be crystallized in complex with JNK1. The HADDOCK model and the crystal structure were similar as shown in Figure 2C, and reassuringly the salt bridges formed as they were designed based on the HADDOCK model (see Fig S3).

Yes, the experiments on Figure 2B summarizes the competitive FP assay data shown on Fig S4, which is now more clearly stated in the figure legend.

2. Overall there is an excessive amount of supplemental data. The authors should consider moving some of this to the main text and removing extraneous material entirely. For example, Figure S3E and the associated legend make two interesting points: that the prior crystal structure of JNK in complex with the ATF2 D-site peptide fragment incorrectly places Lys48 and His49, and also that in the JNK-NFAT4 D-site complex, the peptide backbone overlays well with a helical region of the ATF2 Zn finger. These are key points that should be made in the main text and not buried in the supplement. Another example: Figures S5A/B show deletion/mutagenesis mapping of ATF2 in p38 kinase assays that should be quantified and shown in the main text.

Regarding Fig. S3E: Naturally, the structural analysis is a key aspect of the manuscript, but in order to keep a more general readership engaged in the story that leads up to providing a quantitative picture on how kinase recognition determines the systems level behavior of transcription factor activity, we had to make compromises on how to present the structural details.

Regarding Fig. S5A/B: We believe that the data presented in these panels only supports/supplements what is shown on the different panels of Fig. 3. The in vitro phosphorylation data for the JNK- and p38-binding deficient mutants are shown and quantified on Fig. 3C. The interaction region mapping NMR data on Fig. S3B plotted as I/I₀ is shown as CSP on Fig. 3D.

3. p. 13 1st paragraph, can the authors clarify and provide context for the statement “interestingly, the

interaction is in slow exchange”.

We provided some extra information regarding the observation that the interaction is in slow exchange.

“Interestingly, the interaction is in slow exchange in the NMR timescale - where the chemical shifts for bound and free pp-p38 could be both observed. This is in contrast to other weak MAPK-protein complexes that normally display fast or intermediate exchange, where peaks shift or undergo large intensity changes, respectively. In brief, the doubling of peaks suggests that the ATF2 TAD-bound pp-p38 state is unique and differs from the unbound state.”

4. The quantitative modeling of ATF2 is interesting, but I'm confused about the point being made in panel 7D showing the theoretical impact of perturbing MAPK-ATF2 interaction parameters. The perturbations are pretty extreme – 100x changes in affinities that are probably unrealistic. Also, can the authors comment on the relationship between p38/JNK activity on pathway dynamics or maximal levels of activation? I don't get a sense of this from the way they present the model, but it must provide that information. Finally, could the authors speculate on what cellular conditions would provide differential regulation of p38 vs. JNK?

This panel is meant to show the impact of 10- or 100-fold changes in the binding affinity of the different MAPKs on ATF2 TAD phosphorylation. A 100-fold increase may sound extreme but it is possible to achieve such changes by simple point mutations via creating new H-bonds, salt bridges or hydrophobic contacts at only a few critical positions at the protein-protein interface (regarding JNK:ATF2 TAD binding, see the example of the Q34R/H47R ATF2 double mutant shown on Figure 2B). These scenarios are theoretical but these changes could effectively shape the TAD's transcriptional response in a quantitative way as modeled on this panel.

The 3D plots on Figure 7 give a quantitative picture on TAD phosphorylation as the function of phosphorylated MAPK concentration in the cell. For example, 0.01 on the pp-38 axis on Fig, 7C means that 1% of the total p38 pool (1 μ M in the simulation) is phosphorylated, and 0.1 means 10 %. The pp-JNK axis is similarly scaled. The pp-ATF2 axis is also similarly set up: 1.0 means that 100% of the total ATF2 pool in the cell is phosphorylated. After 40 minutes of stimulation the system is in dynamic equilibrium and the corresponding levels of activated MAPKs and pp-ATF2 levels could be read from the 3D plots. MAPK activation levels were set by the k_{stim6} and k_{stim7} activation constants artificially so that to be able to model different amounts of p38 or JNK activation. Under physiological conditions these MAPKs can get activated to different levels (in a broad range: from a few percent to close to maximum) depending on the strength of the upstream stimuli (e.g. receptor activation followed by MAP3K and MAP2K activation). The dynamics of the response was not addressed in this figure because we wanted to focus on the differences at steady state, namely when S90 switch is ON (WT) or OFF (S90N). Naturally, we can calculate the kinetics of the response, too, and this is shown below for WT ATF2, but we believe that Figure 7 adequately captures the pp-ATF2 response by showing this only after reaching dynamic equilibrium.

The same set of stimuli may elicit different MAPK activation levels depending on the cell type and this is often offset in diseases (e.g. cancer or in chronic inflammation). Thus, p38/JNK activation levels broadly change and the S90 switch will influence pp-ATF2 levels depending on the kinetics and the amplitude of upstream activation. The model in Figure 7 was adjusted to describe MAPK-ATF2 phosphorylation elicited by anisomycin treatment in HEK293T cells, but it can be adjusted to other JNK/p38 activation patterns from other cells. Based on the mechanistic model, we expect that JNK- or p38-pathway activation would affect ATF2 mediated transcription differently depending on the cell-type or the physiological conditions of cells.

Reviewer #2:

Comments on the figures.

Fig. 1. Panel A sequence is too small to read. It needs to highlight all of the amino-acids under discussion including K48. It could also highlight the residues discussed in Fig. 2

Fig. 1A was modified to make the sequence larger.

Fig. 2. B has some lettering that is too small.

The letters on the panel were made bigger.

Comments on the writing.

1. In a couple of places, adding a phrase would make the paper much more accessible.

a. page 7 "We examined JNK1-ATF2 binding by NMR Spectroscopy." The phrase should be added "based on an existing NMR spectral assignment and structure for ATF2 and crystal structure of JNK-peptide interactions."

The text was changed as suggested.

b. page 10 legend "Coelenterazine" should be replaced with "the luciferase substrate Coelenterazine."

The text was changed as suggested.

2. The introduction. The introductory paragraph introduces concepts not known to non-experts, especially Mediator complex. The word "instrumental" implies action, "essential" would be better.

It is described now that the Mediator complex binds to RNA polymerase II and the word usage was improved in the relevant part of the Introduction.

3. The conclusion. The conclusion is over-long. One paragraph that can go concerns prolactin hormone on p 22. The paragraph on Elk-1 (same page) should also go. All of these ideas (self-limiting responses, gene regulatory evolution) have been well discussed in many settings, so the purpose of bringing out these specific examples was lost on me. Best to highlight your first paragraph on p. 23 without these distractions.

The conclusion was shortened and the ELK1 and prolactin hormone related paragraphs were removed as suggested.

Reviewer #3:

Comments:

1. *“The in-cell parameters were determined based on the response of HEK293T cells to anisomycin treatment”*

- The statement is incorrect. For full kinase activity, both T183 and Y185 of the activation loop of JNK must be phosphorylated. JNK is phosphorylated preferentially on Tyr by MKK4 and on Thr by MKK7. There are two main MAPKKs which activate p38, MKK3 and MKK6, and p38 is phosphorylated on T180 and Y182. The model accounts only for a single site (de)phosphorylation of JNK and p38 by MKK7 and MKK6, and the cognate phosphatases. In addition, JNK feedback phosphorylates and further activates MKK7 (doi: 10.1126/scisignal.aab0990). Because the authors do not know the kinetic reaction laws and have not measured the kinetic parameters of (de)phosphorylation, physiological, in-cell activation of JNK and p38 was not modeled.

The word “in-cell” was deleted from this sentence.

The kinetics of MAPK phosphorylation was indeed derived from monitoring the double phosphorylated forms of the MAPKs only (see Figure 7B) and we used a simplified model for processes upstream of MAPKs for activation as well as for dephosphorylation. This has two reasons: 1) we cannot measure parameters that would be required for a more detailed model and 2) we wanted to focus on the processes downstream of the MAPKs. This indeed involved simplification down to an unphosphorylated inactive and a phosphorylated “active” form of the MAPKs.

To show that this pragmatic simplification did not fundamentally affected the outcome, we made a new model where instead of two states (inactive nonphosphorylated and active phosphorylated), MAPKs had 4 states (including a threonine and a tyrosine phosphorylated and double-phosphorylated; which is physiologically more correct) but this did not change pp-ATF2 levels once the system was near to steady-state (which is the case for all panels shown on Figure 7).

We modified the text so that it would be more clear that our model falls short of being complete in several aspects from a real physiological activation/deactivation scenario. However, we believe that this a reasonable decision as including single and double-phosphorylated MAPK states would have greatly increased the number of parameters. We added the following sentence to the description of the quantitative model in the main text:

“In order to focus on MAPK→substrate phosphorylation and keep the number of parameters low, we used a simplified model to implement MAPK activation and dephosphorylation where MAPKs had only two states: an unphosphorylated inactive and a phosphorylated “active”. MAPKs were activated by upstream kinases (k6 or k7) and deactivated by phosphatases (dp1 and dp2) at a single site, and the action of these enzymes on MAPKs were simply modeled as first order reactions. ”

Regarding referencing for MKK6 and MKK7 in the modeling section, the corresponding Methods section, and in the legend for Fig. 7: these concrete kinase names were replaced with a more general reference to k6 and k7 that activate p38 and JNK, respectively.

2. *“Therefore, the model was extended to include upstream kinases and deactivating phosphatases for MAPKs, as well as phosphatases that counteracted MAPK mediated phosphorylation of ATF2.”*

- *The statement has to be tone down. A model does not explicitly consider kinases upstream JNK and p38. The phosphatases in the model are introduced as merely first order reactions.*

This was toned down: Instead of “ Therefore, the model was extended to include upstream kinases and deactivating phosphatases for MAPKs ...”, the text now reads: “Therefore, the model was extended to include the ACTION of upstream kinases and deactivating phosphatases for MAPKs ...”

Please also see the response to Comment 1.

3. *“The mechanistic model was then used to calculate the in-cell phosphorylation level of the ATF2 T69/T71 phosphoswitch (pp-ATF2) under varying amounts of pp-p38 and pp-JNK levels. Double-phosphorylated p38 bound ATF2 stronger than non-phosphorylated p38”*

- *Again, only a single (de)phosphorylation site was modeled for p38 and JNK. Can the authors comment on that in the paper?*

“under varying amounts of pp-p38 and pp-JNK levels“ was changed to “under varying amounts of activated p38 and JNK levels” and also see response to Comment 1. Figure 7 legend states now that “pp-JNK and pp-p38 levels denote the “activated” state of the MAPKs.”

4. *The model describes the binding of p38 to ATF2 that involves two ATF2 sites as the simultaneous binding to both sites. In addition, p38 can bind to ATF2 at only one site, DRS. However mechanistically, in terms of chemistry, the binding to two sites is likely sequential, rather than simultaneous. For instance, following the first binding to DRS, the subsequent binding to the FRS can occur. The rule-based model can readily include all these potential chemical scenarios.*

In the model we indeed accounted only for simultaneous binding to both sites (DRS and FRS) regarding the activated p38-ATF2 complex (two-site binding model). This had a pragmatic reason, the in vitro setup that we used to measure p38→ATF2 phosphorylation gave a k_{cat} value that is relevant to this scenario (see Fig. S7B). We agree that this is a simplification and DRS and FRS binding could also be introduced as two distinct events that can lead to phosphorylation (albeit these two binding events would lead to phosphorylation with distinct catalytic characteristics and correct handling of these would require the determination of new parameters since these were not at hand; independent one-site binding model). The contribution of DRS or FRS based independent binding, however, is likely marginal towards phosphorylation. In order to substantiate this statement, we measured the k_{cat} and K_m of p38 mediated phosphorylation of the ATF2(59-100) construct that lacked the DRS but had an intact FRS (K_m : 51 μ M, k_{cat} : 1.8/s (p-T71); note that the experimentally determined K_m and k_{cat} for the WT construct - DRS+FRS bipartite binding scenario - was 17 μ M and 5.8/s (p-T71), respectively; see Fig. S7B). Similarly, we also attempted to measure the parameters for DRS mediated phosphorylation with another construct that lacked the FRS, however, this reaction was so inefficient that we could not measure its parameters (e.g. K_m was above 1 mM). Despite this, we created a new rule-based model involving independent one-site and simultaneous two-site binding reactions, but the simulation outcome did not fundamentally change compared to the presented model (not shown).

Sequential binding on these two sites is also a formal possibility where binding at one site may influence binding at the other, and thus DRS and FRS binding are not independent (sequential one-site binding model). Technically, however, correctly addressing this would be difficult because the determination of the conversion parameters from an independent to sequential one-site model would be technically not straightforward. However, we believe that this is not necessary and our approach where

we simulated pp-p38→ATF2 phosphorylation based on a simple two-site binding model and its experimentally measurable parameters is a pragmatic solution that does not greatly affect the outcome.

5. In the model a JNK inhibitor does not affect JNK binding to ATF2, while decreasing the JNK catalytic phosphorylation constant to zero. Is this assumption supported by the experimental data?

JNK-IN-8 is a covalent inhibitor attacking a conserved cysteine by the ATP-binding site of JNK1/2/3 (Zhang et al, 2012; <https://doi.org/10.1016/j.chembiol.2011.11.010>). Crystal structure of analogues molecules with JNK3 indicates (PDB ID: 3V6R and 3V6S for JNK-IN-2 and JNK-IN-7, respectively) that covalent binding of the inhibitor does not influence the overall structure of the DRS. The in-cell NanoBit assay can be used to address the effect of inhibitors on protein-protein interactions. We have not observed significant changes on JNK-ATF2 binding after 30 min of JNK-IN-8 treatment in HEK293T cells compared to the untreated cells, although the docking mutant ATF2 (K48E) had diminished binding to JNK (see bar graph below).

We indeed decreased the catalytic constant of JNK to zero in the presence of JNK-IN-8. This assumption stands if all pp-JNK molecules in the cell were covalently bound by JNK-IN-8. Based on the WB blot data shown below this was indeed the case in our experiments. Firstly, JNK-IN-8 covalently attaches to the JNK isoforms causing a small upward mobility shift of JNKs which could be detected (see the pp-JNK blot below). More importantly, in the presence of the inhibitor we did not see any increase of the pp-ATF2 signal upon stimulating HT-MLK3-MKK7 cells by doxycycline (DOX) compared to the basal condition, suggesting that JNK activity was efficiently blocked.

Data on these two figures collectively suggest that JNK-IN-8 efficiently blocked JNK activity without interfering with JNK-ATF2 binding in the cell.

Minor:

1. *“D-motifs are normally located 10-50 amino acids N-terminal,”*

- *Suggested: D-motifs are normally located 10-50 amino acids (FROM THE) N-terminal, ...*

To emphasize the different location of D-motifs and F-motifs, this sentence was rephrased.

2. *“This data was used to create a mechanistic model on cellular TAD phosphorylation ...”*

- *Suggested: This data was used to create a mechanistic model OF cellular TAD phosphorylation ...*

The text was changed as suggested.

3. *“The transactivating region of both cJUN or ATF2 contains a MAPK binding D-motif”*

- *Suggested: The transactivating region of both cJUN AND ATF2 contains a MAPK binding D-motif*

The text was changed as suggested.

4. *“Next, we assessed the role of JNK binding on MAPK target site phosphorylation. The Znfinger+D-motif module (19-58) was left intact or replaced by cJUN or NFAT4 D-motifs and phosphorylation of the 69-TPTP-72 target site by activated JNK was monitored by anti-phospho immunoblots (Figure 1E). This analysis confirmed the importance of K48 and K35 in ATF2 and cJUN phosphorylation, respectively.”*

- *Please clarify this conclusion, because a large Znfinger+D-motif module and not only K48 or K35 was substituted by other motifs.*

The texts was replaced with the following:

“Next, we assessed the role of JNK binding on MAPK target site phosphorylation. The Zn-finger+D-motif module (19-58) was left intact or mutated (K48E) or replaced by cJUN or its mutated version (K35E) and phosphorylation of the 69-TPTP-72 target site by activated JNK was monitored by anti-phospho immunoblots (Figure 1E). This analysis confirmed the importance of key lysine residues in ATF2 Zn-finger+D-motif or cJUN D-motif mediated phosphorylation.”

REVIEWERS' COMMENTS

Reviewer #1 (Remarks to the Author):

The authors have addressed all of my concerns from the original review in their revised manuscript.

Reviewer #3 (Remarks to the Author):

The rebuttal is well written, and the minor editing of the text is nicely done. The presented model is basically the same as before, as the author who has developed the model knows well. The incorporation of the second binding site is done in a way that the affinity for one binding site is extremely small and practically all the p38 binding occurs at a single site or simultaneously, as the authors mentioned in the rebuttal. Because the presented model is an illustration of the experimental findings rather than a main result of this work, this reviewer does not have further comments and do not object to the acceptance of this manuscript.